# Brn3a controls the soma localization and axonal extension patterns of developing spinal dorsal horn neurons

**Kazuhiko Nishida**[1]*, **Shinji Matsumura**[1], **Hitoshi Uchida**[2¤], **Manabu Abe**[2,3], **Kenji Sakimura**[2,3], **Tudor Constantin Badea**[4,5], **Takuya Kobayashi**[1]

**1** Department of Medical Chemistry, Kansai Medical University, Hirakata, Osaka, Japan, **2** Department of Cellular Neurobiology, Brain Research Institute, Niigata University, Niigata, Japan, **3** Department of Animal Model Development, Brain Research Institute, Niigata University, Niigata, Japan, **4** Research and Development Institute, Faculty of Medicine, Transylvania University of Brasov, Brasov, Romania, **5** National Brain Research Center, ICIA, Romanian Academy, Bucharest, Romania

¤ Current address: Department of System Pathology for Neurological Disorders, Brain Research Institute, Niigata University, Niigata, Japan

* nishidka@hirakata.kmu.ac.jp

**Data Availability Statement:** All relevant data are within the paper and its Supporting Information files.

## Abstract

The spinal dorsal horn comprises heterogeneous neuronal populations, that interconnect with one another to form neural circuits modulating various types of sensory information. Decades of evidence has revealed that transcription factors expressed in each neuronal progenitor subclass play pivotal roles in the cell fate specification of spinal dorsal horn neurons. However, the development of subtypes of these neurons is not fully understood in more detail as yet and warrants the investigation of additional transcription factors. In the present study, we examined the involvement of the POU domain-containing transcription factor Brn3a in the development of spinal dorsal horn neurons. Analyses of Brn3a expression in the developing spinal dorsal horn neurons in mice demonstrated that the majority of the Brn3a-lineage neurons ceased Brn3a expression during embryonic stages (Brn3a-transient neurons), whereas a limited population of them continued to express Brn3a at high levels after E18.5 (Brn3a-persistent neurons). Loss of Brn3a disrupted the localization pattern of Brn3a-persistent neurons, indicating a critical role of this transcription factor in the development of these neurons. In contrast, Brn3a overexpression in Brn3a-transient neurons directed their localization in a manner similar to that in Brn3a-persistent neurons. Moreover, Brn3a-overexpressing neurons exhibited increased axonal extension to the ventral and ventrolateral funiculi, where the axonal tracts of Brn3a-persistent neurons reside. These results suggest that Brn3a controls the soma localization and axonal extension patterns of Brn3a-persistent spinal dorsal horn neurons.

## Introduction

The spinal dorsal horn (SDH) plays a crucial role in the transmission and modulation of sensory information. It comprises various neurons that exhibit distinctive cell body localization,

**Funding:** This work was supported by Japan Society for the Promotion of Science (JSPS) KAKENHI Grant Number JP 19K07855, JP 22K07383, JP 16H06276 (AdAMS), and Unitatea Executiva pentru Finantarea Invatamantului Superior, a Cercetarii, si Inovarii Grant Number PN-III-P4-PCE-2021-0333. The funders had no role in study design, data collection and analysis, decision to publish, or preparation of the manuscript.

**Competing interests:** The authors have declared that no competing interests exist.

morphology, and neurotransmitter expression [1]. During mouse development, spinal dorsal horn neurons originate from a specific progenitor domain located along the ventricular zone of the neural tube from E10 to E12.5. Six early-born (dI1-dI6) and two late-born (dIL$_A$ and dIL$_B$) progenitor populations have been well-characterized based on the localization and expression patterns of transcription factors [2]. Moreover, ample evidence indicates that transcription factors expressed in a specific progenitor domain largely determine the identity of each subset of spinal dorsal horn neurons. For instance, regarding the neurotransmitter identity of superficial dorsal horn neurons, Tlx3 expressed in the dI5/dIL$_B$ domain plays a crucial role in the specification of excitatory cell fate, whereas Ptf1a controls the cell fate of the dI4/dIL$_A$ population toward an inhibitory fate. However, the extent of heterogeneity of spinal dorsal horn neurons is clearly far beyond that conventionally defined by progenitor zones. For instance, the dI1 population produces two neuronal subtypes that differ in their ipsilateral and contralateral axonal projections [3]. In addition, the dI5/dIL$_B$ population includes neurons with completely different axonal morphologies: some are locally projecting, whereas the others are supraspinal projection neurons [4]. In the latter example, Phox2a, a homeobox transcription factor, was recently shown to be expressed in a subpopulation of dI5/dIL$_B$ progenitors and to regulate the development of supraspinal projection neurons [5]. Thus, the investigation of additional transcription factors is imperative to uncover the developmental program underlying the diversification of spinal dorsal horn neurons.

POU-type IV proteins constitute a family of homeobox-containing transcription factors that are widely conserved across the animal phylogeny [6]. This family contains three members in mammals, namely Brn3a, Brn3b, and Brn3c (also called POU4f1, POU4f2, and POU4f3, respectively), which exhibit partially overlapping expression patterns in the subsets of developing neurons in the central and peripheral nervous systems. POU family members expressed in each region of the nervous system play distinct roles in neural development, such as survival, dendritic and axonal morphology formation, and migration [6]. Brn3a, and to a lesser extent Brn3b and Brn3c, are expressed in developing spinal dorsal horn neurons [7, 8]. We previously showed that a population of these neurons continues to express Brn3a in adults and is localized in two discrete regions of the dorsal horn: the marginal and deeper laminae [9]. Although Brn3a expression in the developing spinal dorsal horn is well known, the identity of Brn3a-positive spinal dorsal horn neurons and the significance of Brn3a in their development have not been extensively studied. We recently demonstrated that Brn3a-positive spinal dorsal horn neurons are localized to the marginal laminae including supraspinal projection neurons, and are involved in visceral pain transmission [9]. Zou et al. reported that Brn3a-knockout (KO) mice did not exhibit defects in the overall distribution of excitatory neurons [10]. However, their study did not directly clarify the effect of Brn3a deficiency on the cell fate of Brn3a-positive neurons.

In the present study, we analyzed Brn3a expression in the developing spinal dorsal horn and explored the effects of Brn3a deficiency and overexpression on the development of spinal dorsal horn neurons. Our findings suggest the involvement of Brn3a in regulating soma localization and axonal extension in a population of spinal dorsal horn neurons.

## Materials and methods

### Animals

C57BL/6J mice were purchased from CLEA Japan, Inc. (Tokyo, Japan). Brn3a-cKOAP [11] and Ai9 reporter [12] mice were obtained from Jackson Laboratory (010558, 007909; Bar Harbor, ME, USA). Brn3a-Cre knock-in mice were generated as described previously [13]. To generate Brn3a$^{+/-}$ mice, Lbx1$^{Cre/+}$;Brn3a$^{cKOAP/+}$ female mice were used because we empirically

know that Cre-mediated recombination occurs in the female germline. Brn3a$^{cKOAP/cKOAP}$ mice were crossed with Lbx1-Cre mice [14] to generate Lbx1$^{Cre/+}$;Brn3a$^{cKOAP/+}$ mice, and Brn3a$^{AP/+}$ (Brn3a$^{-/+}$) mice were screened from the offspring of Lbx1$^{Cre/+}$;Brn3a$^{cKOAP/+}$ female mice. Brn3b-CreERT mice were generated using the embryonic stem (ES) cell line RENKA, which is derived from the C57BL/6N strain (S1 Fig) [15]. Homologous recombinants among ES cells were identified by Southern blotting. The FRT-flanked neomycin resistance gene cassette was removed by crossing the mice with the Flippase-driver line (Actb-Flpe) [16]. For the cell fate analysis of Brn3b-lineage neurons, 250 µL of corn oil (C8267; Sigma, St. Louis, MO, USA) containing 1 mg of tamoxifen (T5648; Sigma) was intraperitoneally injected into pregnant Brn3b$^{CreERT/+}$;Ai9 mice once daily from E10.5 to E14.5. The mice were maintained on a 12–12 h (8:00 to 20:00) light-dark schedule. All animal experiments were conducted in accordance with the Japanese guidelines and regulations (2006) for scientific and ethical animal experimentation and were approved by the Animal Experimentation Committee of Kansai Medical University (22–056, 22–057).

## Vectors

*pCAG*, *pCAG-Cre*, *pCAG-mCherry*, *pCAG-nlsEGFP*, *pCAG-LSL-EGFP*, *pCAG-Brn3a*, and *pEF-RL* were generously gifted by Drs. Junichi Miyazaki, Yan Zhu, Yasuto Tanabe, Kenta Yamauchi, Fujio Murakami, Eric Turner, and Hiroshi Itoh, respectively [17–22]. *pGEMT-pre-prodynorphin* (*Pdyn*) and *pGEMT-cholecystokinin* (*CCK*) are described previously [9]. To obtain the *pCAG-nlsEGFP-CAG-Brn3a* vector, the *EYFP* gene in the *pCAG-EYFP-CAG* vector (a generous gift from Dr. Tetsuichiro Saito [23]) was replaced by the *nlsEGFP* gene, followed by the insertion of *Brn3a* into the multi-cloning site of the vector. To obtain the *pCAG-Brn3b*, *pCAG-FLAG-Brn3a-POU*, and *pCAG-FLAG-Brn3a-ΔPOU* constructs, cDNA for mouse Brn3b (249–1484 of NM_138944.3), FLAG-tagged POU domain of mouse Brn3a (1050–1532 of AY706205.1), and FLAG-tagged mouse Brn3a lacking the POU domain (267–1331 of AY706205.1) were PCR-amplified and subcloned into the *pCAG-MCS* vector [20]. To obtain the *pGL2B-TrkA enhancer* construct, the 5'-flanking region of mouse *TrkA* gene (79799–80637 of AC161454.3) was PCR-amplified from the genomic DNA of C57BL6J mice and subcloned into the *pGL2-basic* vector (Promega, Fitchburg, WI, USA). The following primers were used for PCR amplification: (Brn3b forward) 5′-GAT CGG TAC CAT GAT GAT GAT GTC CCT G-3′, (Brn3b reverse) 5′-ATA TGG TAC CCT AAA TGC CGG CAG AG-3′, (FLAG-Brn3a-POU forward) 5′-ATA TGG TAC CAT GGA CTA CAA GGA CGA CGA TGA CAA GGA CTC GGA CAC GGA CCC G-3′, (FLAG-Brn3a-POU reverse) 5′-GAT CGG TAC CTC AGT AAG TGG CAG AG-3′, (FLAG-Brn3a-ΔPOU forward) 5′-GAT CGG TAC CAT GGA CTA CAA GGA CGA CGA TGA CAA GAT GAT GTC CAT GAA CAG C-3′, (FLAG-Brn3a-ΔPOU reverse) 5′-ATA TGG TAC CTC AGC CGC CGT TGA AGA GCT C-3′, (pGL2B-TrkA forward) 5′-GCG CGG ATC CAC TAA GAG ATC TAT TAA TTT CTC CGC-3′, (pGL2B-TrkA reverse) 5′-ATA TGG ATC CGG CGC CCG CCT TGC TCG-3′.

## Immunohistochemistry and *in situ* hybridization

Embryonic mice from E14.5 to E18.5 were transcardially perfused with 4% paraformaldehyde solution (0.1 M phosphate buffer [pH 7.4] containing 4% paraformaldehyde) whereas those at E12.5 were dissected without transcardial perfusion. Spinal cords at E16.5 and E18.5 were then peeled off from the vertebral column whereas those from E12.5 and E14.5 were kept with it in order not to damage the sample during dissection. The spinal cord samples were immersed in the same fixative for 3 h, followed by incubation in 0.1 M PBS containing 30% (w/w) sucrose

overnight at 4˚C. The samples were then embedded in the OCT compound and stored at −80˚C until further examination.

Transverse sections of the thoracic and lumbar spinal cords of mice were prepared using a cryostat (20 μm sections). Immunohistochemistry and *in situ* hybridization were performed as previously described [9]. Antibodies used in the present study were mouse anti-Brn3a (1:300, Sigma, MAB1585), rabbit anti-Brn3b (1:500, generous gift from Dr. Eric Turner [24]), sheep anti-Zfhx3 (1:50, R&D Systems, Minneapolis, MN, USA, AF7384-SP), rabbit anti-CGRP (1:2000, Sigma, C8198), mouse anti-FLAG (1:500, Medical Chemistry Pharmaceutical, Co., Kobe, Japan, K0602-S), Alexa 488-conjugated donkey anti-sheep IgG (1:300, Invitrogen, Carlsbad, CA, USA, A11015), Alexa 488-conjugated donkey anti-mouse IgG (1:300, Invitrogen, A21202), Alexa 488-conjugated donkey anti-rabbit IgG (1:300, Invitrogen, A21206), Alexa 488-conjugated goat anti-mouse IgG (1:300, Invitrogen, A11029), Alexa 594-conjugated goat anti-rabbit IgG (1:300, Invitrogen, A11037), Alexa 594-conjugated goat anti-mouse IgG (1:300, Invitrogen, A11032), and Cy3-conjugated donkey anti-mouse IgG (1:300, Jackson ImmunoResearch, West Grove, PA, USA, 715-165-150).

The sections were counterstained with Hoechst 33342 (1:2,000, Thermo Fisher Scientific Inc., H3570) and mounted with a mounting reagent containing 25% glycerol, 0.1 M Tris (pH 8.5), 27.5 mg/mL 1,4-Diazabicyclo[2.2.2]octane (DABCO, Tokyo Chemical Industry Co., Tokyo, Japan, D0134), and 100 mg/mL Mowiol 4–88 (Merck, Darmstadt, Germany, 475904). Fluorescence images were obtained using a confocal microscope (LSM700; Carl Zeiss, Jena, Germany; Dragonfly; Oxford Instruments, Abington-on-Thames, UK).

## *In utero* electroporation

*In utero* electroporation was performed as described previously [25]. Briefly, pregnant mice at E11.8 or E12.5 were deeply anesthetized with a mixture of medetomidine (37.5 μg/kg, Nippon Zenyaku Kogyo, Fukushima, Japan), midazolam (2 mg/kg, Sandoz, Tokyo, Japan), and butorphanol (0.25 mg/kg, Meiji Seika Pharma, Tokyo, Japan) prior to electroporation. Plasmid DNA was introduced into the central canal of the spinal cord of embryos using a microinjector (IM-31; Narishige, Tokyo, Japan). Round electrodes (CUY650P2, CUY650P0.5; Nepagene, Ichikawa, Japan) were attached to the uterus, and five electric pulses (30–35 V, 50 ms) were applied using an electroporator (CUY21SC; Nepagene). For gene transfer into Brn3a$^{cKOAP/+}$ and Brn3a$^{cKOAP/cKOAP}$ mice, 0.05 mg/mL of *pCAG-Cre*, together with 0.4 mg/ml of *pCAG-nlsEGFP* or 0.4 mg/mL of *pCAG-nlsEGFP-CAG-Brn3a*, were introduced into the neural tube at E11.8. In the case of rescue experiments with Brn3b and Brn3a-POU, 0.05 mg/mL of *pCAG-Cre* and 0.4 mg/mL of *pCAG-nlsEGFP*, together with 0.4 mg/mL of *pCAG-Brn3b* or 0.4 mg/mL of *pCAG-FLAG-Brn3a-POU*, were introduced. For gene transfer into Brn3a$^{Cre/+}$ mice, 0.2 mg/mL of *pCAG-LSL-EGFP* with or without 0.2 mg/mL of *pCAG-Brn3a* were introduced into the neural tube at E12.5.

## Alkaline phosphatase (AP) assay

The AP reaction on spinal cord sections was performed as previously described [26], with some modifications. Brn3a$^{cKOAP/+}$ and Brn3a$^{cKOAP/cKOAP}$ mice at E18.5 were transcardially perfused with 0.1 M phosphate buffer (pH 7.4) containing 2% paraformaldehyde, and their spinal cords were dissected. The spinal cord samples were post-fixed with the same fixative for 3 h, followed by incubation in 0.1 M PBS containing 30% (w/w) sucrose overnight at 4˚C. The samples were then embedded in the OCT compound and stored at −80˚C until further examination.

Transverse sections of the thoracic spinal cords (20 μm in thickness) were post-fixed with the same fixative for 10 min, washed three times with PBS, and mounted with a mounting

reagent described above. The fluorescence of nlsEGFP in the sections was observed under a confocal microscope (LSM700). The sections were then incubated in PBS to remove cover glasses, washed with PBS containing 0.2% Triton X-100 three times, transferred to PBS, and heated in a water bath for 1 h at 65˚C to inactivate endogenous AP activity. After washing the sections with AP buffer (0.1M Tris [pH 9.5], 0.1M NaCl, 50 mM $MgCl_2$, and 0.01% Tween-20) three times, AP reaction was performed in AP buffer containing 0.33 mg/mL nitroblue tetrazolium (NBT; FUJIFILM Wako, Osaka, Japan), and 0.17 mg/mL 5-bromo-4-chloro-3-indolyl-phosphate (BCIP; Roche, Basel, Switzerland) at room temperature for 1 h (Brn3a$^{cKOAP/cKOAP}$ mice) or 2 h (Brn3a$^{cKOAP/cKOAP}$ mice). To detect AP-positive fibers, the AP reaction was performed for 2 h (Brn3a$^{cKOAP/cKOAP}$ mice) or 4 h (Brn3a$^{cKOAP/cKOAP}$ mice). After the AP reaction, the sections were washed two times with PBS, post-fixed with 4% paraformaldehyde solution for 15 min, and mounted with a mounting reagent. Images of the sections were captured using a confocal microscope (LSM700) in the same frame as the nlsEGFP photographs obtained previously. The localization of AP- and nlsEGFP-double-positive cells was analyzed using ImageJ.

## Luciferase reporter assay

Luciferase reporter assay was performed according to the manufacturer's protocol using the Dual Luciferase Reporter Assay System (Promega, E1910). For transfection, 0.2 μg of *pGL2-basic* vector constructs and 0.001 μg of *pEF-RL* together with 1.2 μg of *pCAG*, *pCAG-Brn3a*, *pCAG-Brn3a-POU*, or *pCAG-Brn3a-ΔPOU* were introduced using Lipofectamine (Invitrogen) into COS7 cells cultured in 35 mm dishes. After 48 h, the activity of firefly and *Renilla* luciferase in the cell lysates was measured using a luminometer (AB-2200; ATTO Co., Tokyo, Japan). The activity of firefly luciferase was normalized to that of *Renilla* luciferace, and the fold change in activity relative to the control sample (*pGL2-basic* plus *pCAG*) was calculated.

## AAV injection

AAV carrying the *tdTomato* and *SypEGFP* genes [27] was a generous gift from Dr. Hongkui Zeng (Addgene, #51509-AAV1; Watertown, MA, USA). Brn3a$^{Cre/+}$ mice (2–3 months of age) were deeply anesthetized with a mixture of medetomidine (37.5 μg/kg, Nippon Zenyaku Kogyo), midazolam (2 mg/kg, Sandoz), and butorphanol (0.25 mg/kg, Meiji Seika Pharma), and their backs were shaved using an electric shaver. The intervertebral spaces rostral to the L1 vertebra, which correspond to the L5 spinal segment [28], were exposed by carefully removing the connective tissues. The AAV injection tool included a fine glass capillary (G-1; Narishige) and a Hamilton syringe (80330-701RN; Hamilton Co. Reno, NV, USA) connected by an RN compression fitting (55750–01; Hamilton Co.). The AAV solution was loaded into the glass capillary and injected into three different regions of the L5 spinal dorsal horn on the right side using a microinjector (IMS-20, Narishige). For each injection, 300 μL of AAV solution was used. Following injection, the glass capillary was maintained at the injection site for 3 min to minimize leakage. The wounds were sutured immediately after the operation, and the mice were placed on a heating pad until recovery. Three weeks post operation, the mice were transcardially perfused with 4% paraformaldehyde solution to obtain brain and spinal cord samples.

## Birthdate analysis

For birthdate analysis of Brn3a-persistent neurons, 5-ethynyl-2'-deoxyuridine (EdU, 12.5 mg/kg, Thermo Fisher Scientific Inc. Waltham, MA, USA) was intraperitoneally injected into pregnant C57BL/6 mice once a day (12 P.M.) at E11.5 or E12.5, and the spinal cords of the

mice were dissected at P21. For double labeling of Brn3a and EdU, transverse sections of the thoracic spinal cord of mice were first incubated with an anti-Brn3a, followed by Alexa 594-conjugated goat anti-mouse IgG. Then, EdU was visualized by incubation in 0.1M Tris buffer (pH 8.5) containing 4 mM $CuSO_4$, 11.3 μM Alexa 488-Azide (A10266; Invitrogen), and 0.1 M ascorbic acid.

## Image analysis

For the analysis of Brn3a- and Brn3b-positive neurons at E12.5 and E18.5, fluorescence images of Brn3a- and Brn3b-immunostaining were captured using DragonFly and analyzed using Imaris (version 8.4.1, Oxford Instruments, Abington-on-Thames, UK). Neurons with the mean signal intensity exceeding the threshold (Brn3a: >150, Brn3b: >120) were defined as Brn3a- or Brn3b-positive neurons.

To analyze the localization of AP-positive neurons, both AP and nlsEGFP images (captured by LSM700) were first thresholded using ImageJ (AP: >150, nlsEGFP: >50). Cells whose sizes in the AP- and nlsEGFP-positive regons were >15 pixels were defined as AP- and nlsEGFP-double-positive cells.

To analyze the soma localization of control and Brn3a-overexpressing neurons, images of EGFP (captured by DragonFly) were z-stacked, and the EGFP intensity was thresholded (>175) using ImageJ. Cells whose size was over 200 pixels and circularity of whose EGFP-positive region was between 0.3 and 1.0 were defined as EGFP-positive cells.

To analyze axonal extension derived from control and Brn3a-overexpressing neurons, EGFP-positive axons in the dorsal (DF), ventrolateral (VLF), and ventral (VF) funiculi were analyzed using Imaris. VF and VLF were defined as the white matter ventral to the top of the central canal. The boundary between the gray and white matter was determined using Hoechst 33342 staining. The EGFP signal (captured by DragonFly) was thresholded (>5,000), and the number of EGFP-positive fibers in each region (as defined by their shape; BoundingBoxOO length A > 10 μm, B < 5 μm, C < 5 μm) was determined.

To analyze Brn3a-positive neurons among Brn3a-lineage neurons, *pCAG-LSL-EGFP* was introduced into the spinal cord of Brn3a[Cre/+] mice, and the sections were immunostained with anti-Brn3a. Fluorescence images captured with DragonFly were analyzed using Imaris. The EGFP signal was thresholded (>10,000) and the cells whose "quality" and "number of voxels" were above 2,000 and 1,000, respectively, were defined as EGFP-positive cells. EGFP-positive cells whose mean signal intensity of Brn3a was two times the background level were regarded EGFP- and Brn3a-positive cells.

For the analysis of Brn3b-positive neurons among Brn3b-lineage neurons, sections from Brn3b[CreERT/+];Ai9 mice administered tamoxifen were immunostained with anti-Brn3b, and fluorescence images were captured using LSM700. The intensity of tdTomato and Brn3b signals was thresholded using Imaris (tdTomato > 5,000, Brn3b > 4,000), and tdTomato- and Brn3b-double positive cells were analyzed using the "Spot" tool.

To analyze Zfhx3- and Brn3a-double positive cells, fluorescence images of sections immunostained with anti-Zfhx3 and anti-Brn3a were captured using DragonFly. The intensity of Zfhx3 and Brn3a signals was thresholded by Imaris (Zfhx3 >180, Brn3a >150), and Zfhx3- and Brn3a-double positive cells were analyzed using the "Spot" tool.

For the analysis of Zfhx3-positive cells in Brn3a[+/+] and Brn3a[-/-] mice, fluorescence images of sections immunostained with anti-Zfhx3 were captured using DragonFly and Zfhx3-positive cells were analyzed using Imaris. Zfhx3 signal was thresholded (>40), and cells whose "quality" and "number of voxels" parameters exceeded 25 and 2,000, respectively, were defined as Zfhx3-positive cells.

For birthdate analysis of Brn3a-persistent neurons, fluorescence images of EdU and Brn3a were captured using LSM700, and analyzed using Imaris. Brn3a signal was thresholded (>10,000), and the cells whose "intensity mean" and "number of voxels" parameters exceeded 15,000 and 300, respectively, were defined as Brn3a-positive cells. Brn3a-positive cells with the maximum EdU intensity of >15,000 were defined as Brn3a- and EdU-double-positive cells.

## Statistical analysis

Histochemistry results were analyzed using the Mann–Whitney U-test and Kruskal–Wallis tests followed by the two-stage step-up method of Benjamini, Krieger and Yekutieli to determine the false discovery rate. All statistical analyses were performed using GraphPad Prism 7.02 (GraphPad Software Inc., La Jolla, CA, USA). Data are expressed as mean ± SEM.

## Results

### Expression pattern of Brn3a in the developing spinal dorsal horn

Previous studies have demonstrated that Brn3a is expressed in almost all early post-mitotic progenitor domains giving rise to excitatory dorsal horn neurons [29, 30]. Fewer neurons appear to express Brn3a at the perinatal [31] and adult [9] stages. Brn3a-positive neurons in adults reside in two discrete regions, the marginal and deeper dorsal horns. To elucidate the formation of this localization pattern during embryonic development, we first compared the expression pattern of Brn3a at different embryonic stages. Immunostaining for Brn3a in the E12.5 spinal cord demonstrated that Brn3a was expressed in the mantle layer of the dorsal neural tube (Fig 1A), as previously reported [10]. At E14.5, the Brn3a-void region appeared in the most superficial region of the dorsal horn (Fig 1B). This void region widened at E16.5 and E18.5, in which Brn3a-positive neurons were sparsely distributed (Fig 1C and 1D). At E18.5, the accumulation of Brn3a-positive neurons in the marginal region was apparent (arrows in Fig 1D). The number of Brn3a-positive neurons at E18.5 was much lower than that at E12.5 (E12.5:494.3 ± 41.3 per section [$n$ = 6 mice] and E18.5:109.3 ± 19.3 per section [$n$ = 6 mice]) (Fig 1K). The reduction in the number of Brn3a-positive neurons between E12.5 and E18.5 can be attributed to Brn3a downregulation during embryonic stages. To directly confirm this possibility, we labeled Brn3a-lineage neurons by introducing a Cre-dependent EGFP expression vector (*pCAG-LSL-EGFP*) into Brn3a-Cre knock-in mice [13] via *in utero* electroporation at E12.5 (Fig 2A). This method enables spinal dorsal horn-specific gene delivery (S2 Fig). Analysis of Brn3a-lineage neurons at E18.5 demonstrated that most of them were localized in the Brn3a-void region of the spinal dorsal horn, and, as expected, only a small population of these neurons was Brn3a-positive (13.5 ± 2.5%, 619 cells [$n$ = 6 mice]) (Fig 2B–2D).

In the spinal cord, Brn3b is expressed in developing spinal dorsal horn neurons [10], whereas Brn3c expression is very low [8]. Thus, we quantitatively analyzed Brn3a and Brn3b co-expression to estimate the possible functional redundancy of these molecules. Immunostaining for Brn3a and Brn3b was performed on spinal cord sections at E12.5 and E18.5 (Fig 1E–1J), and the percentage of double-positive neurons among Brn3a-positive neurons at these stages was quantified. The number of Brn3b-positive neurons was much lower than that of Brn3a-positive neurons at both E12.5 and E18.5. Similar to the number of Brn3a-positive neurons, the number of Brn3b-positive neurons also decreased during embryonic development, as confirmed by lineage analysis using Brn3b-CreERT knock-in mice (S1 and S3 Figs). Brn3a- and Brn3b-double positive neurons accounted for 10.2 ± 1.0% of Brn3a-positive neurons at E12.5, and their proportion decreased to 2.2 ± 0.7% at E18.5 (Fig 1L). These results suggest that only a minor population of Brn3a-positive spinal dorsal horn neurons continuously co-

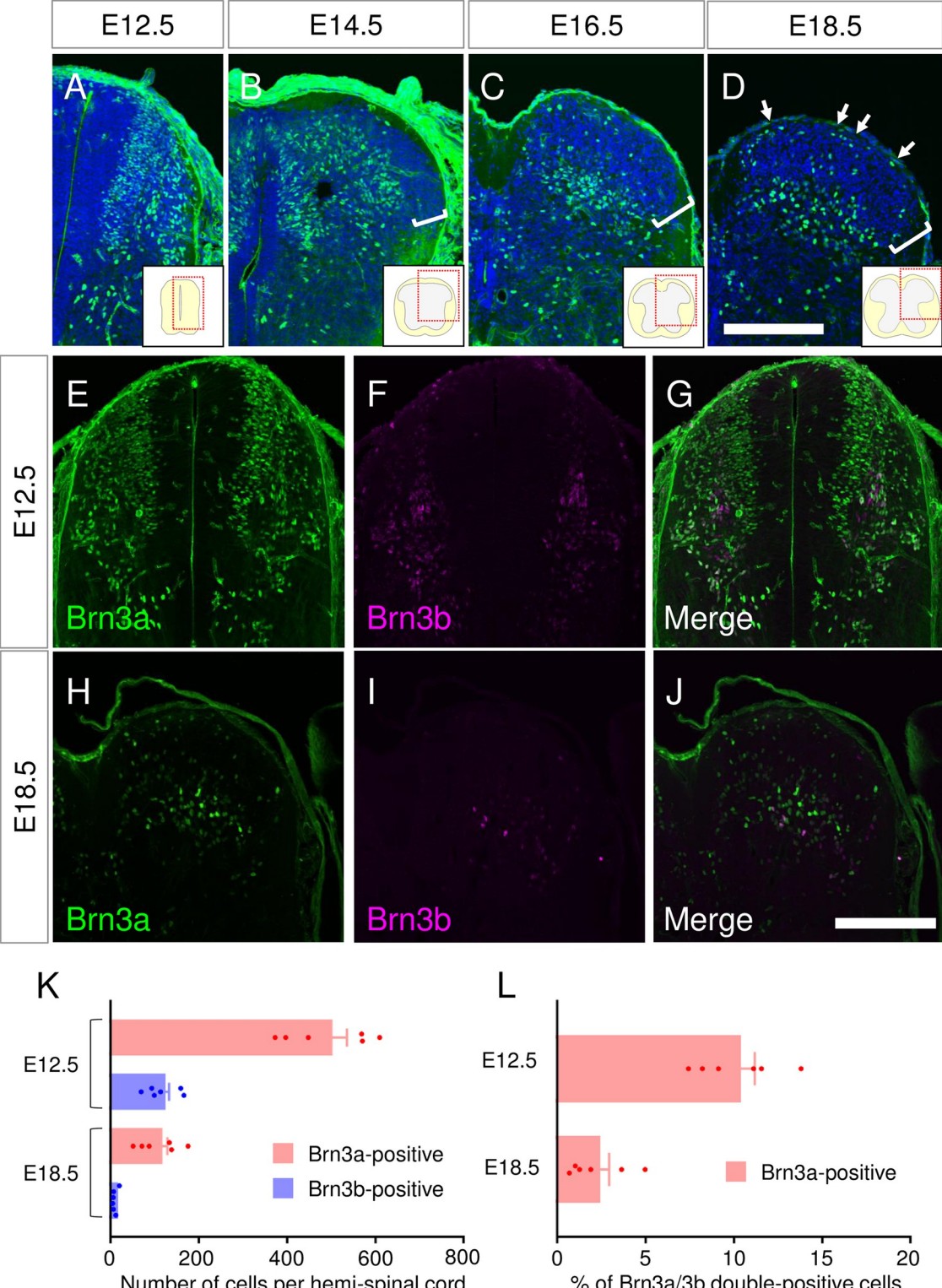

**Fig 1. Expression of Brn3a in the spinal cord at embryonic stages.** (A-D) Immunostaining of Brn3a (green) was performed on the transverse sections of the thoracic spinal cord from C57BL/6J mice at E12.5 (A), E14.5 (B), E16.5 (C), and E18.5 (D). The sections were counterstained by Hoechst 33342 (blue). Brn3a-void region in the dorsal horn is marked by square brackets. Spinal cords on the right side are shown. Arrows in D indicate Brn3a-positive neurons in the marginal region. Schematic diagrams of the spinal cord are shown in the insets. Red dotted squares indicate the area shown in the image. (E-J) Immunostaining of Brn3a (green), together

with Brn3b (magenta), was performed on the transverse sections of the thoracic spinal cord from C57BL/6J mice at E12.5 (E-G) and E18.5 (H-J). Scale, 200 μm. (K) The number of Brn3a (red)- and Brn3b (blue)-positive cells in the hemi-spinal cord at E12.5 and E18.5 is shown. (L) The percentage of Brn3a- and Brn3b-double positive cells among Brn3a-positive cells at E12.5 and E18.5 is shown. Data are presented as the mean ± SEM.

expresses Brn3b during development. Therefore, we focused on Brn3a function for the remainder of the study.

## Loss of Brn3a disrupts the localization pattern of Brn3a-persistent neurons in the spinal dorsal horn

We next focused on the development of a population of Brn3a-positive neurons that continued to express Brn3a after E18.5 (hereafter, "Brn3a-persistent neurons"). To examine the effect of Brn3a deficiency on the development of Brn3a-persistent neurons, we used Brn3a-cKOAP knock-in mice [11]. This mouse enabled us to visualize Brn3a knockout cells by expressing the AP reporter under the control of the Brn3a promoter following Cre-mediated recombination. For the spinal cord-specific introduction of Cre, we performed gene transfer via *in utero* electroporation (Fig 3A). The Cre expression vector was first introduced into the spinal cord of Brn3a$^{cKOAP/+}$ mice at E11.8, and the distribution of AP-positive cells (Brn3a$^{AP/+}$) in these mice was analyzed based on the enzymatic reaction of AP on spinal cord sections at E18.5. Under these conditions, AP-positive (Brn3a$^{AP/+}$) cells were detected around the deeper and marginal dorsal horn (arrows in Fig 3B), resembling the Brn3a expression pattern in the wild-type mice at E18.5 (Fig 1D). AP-positive cells were noted in Brn3a$^{cKOAP/cKOAP}$ mice with Cre introduction (Brn3a$^{AP/AP}$), while they were absent in the marginal region (Fig 3C). The expression of Cre, together with Brn3a or Brn3b, rescued the localization of Brn3a$^{AP/AP}$ cells to the marginal region (Fig 3D and 3E). To quantify the differential distribution of AP-positive cells under these conditions, we sparsely labeled AP-positive cells by introducing a small amount of the Cre expression vector, together with EGFP fused to the nuclear localization signal (nlsEGFP), to distinguish the location of the nuclei of AP-positive cells (S4 Fig). Next, we defined five zones in the dorsal horn at E18.5 based on their distance from the dorsal surface (Fig 3G). Analysis of several marker molecules demonstrated that each zone exhibited differential expression of these markers (S5 Fig), similar to the laminar structure observed in adults [32]. Quantitative analysis of the localization of AP-positive cells (Fig 3B–3E) revealed that the percentage of Brn3a$^{AP/AP}$ cells in zone 1 and 2 was significantly lower than that of Brn3a$^{AP/+}$ cells, whereas the percentage of Brn3a$^{AP/AP}$ cells in zone 5 was significantly higher than that of Brn3a$^{AP/+}$ cells ([Brn3a$^{AP/+}$ vs Brn3a$^{AP/AP}$ in zone 1] $q = 0.0007$; [Brn3a$^{AP/+}$ vs Brn3a$^{AP/AP}$ in zone 2] $q = 0.0003$; [Brn3a$^{AP/+}$ vs Brn3a$^{AP/AP}$ in zone 5] $q = 0.0264$, Kruskal–Wallis test, followed by the Benjamini, Krieger, and Yekutieli method; Fig 3H, S1 File). These results indicate that loss of Brn3a disrupts the localization pattern of Brn3a-persistent neurons in the spinal dorsal horn. Meanwhile, the percentage of Brn3a$^{AP/AP}$ cells overexpressing with Brn3a (Brn3a$^{AP/AP}$+Brn3a) and Brn3b (Brn3a$^{AP/AP}$+Brn3b) in zone 1 and 2 was significantly higher than that of Brn3a$^{AP/AP}$ cells ([Brn3a$^{AP/AP}$ vs Brn3a$^{AP/AP}$+Brn3a in zone 1] $q = 0.0088$; [Brn3a$^{AP/AP}$ vs Brn3a$^{AP/AP}$+Brn3a in zone 2] $q = 0.0109$; [Brn3a$^{AP/AP}$ vs Brn3a$^{AP/AP}$+Brn3b in zone 1] $q = 0.0094$, Kruskal–Wallis test, followed by the Benjamini, Krieger, and Yekutieli method; Fig 3H, S1 File), indicating that Brn3a and Brn3b overexpression rescued the phenotype of Brn3a deficiency. In addition, the reduction in the number of Brn3a-KO cells in the marginal zone was analyzed by using Brn3a$^{-/-}$ mice. Zfhx3, a homeobox transcription factor, is expressed in the marginal lamina at E18.5 [33], a population of which was Brn3a-positive (10.1 ± 1.8% [$n = 6$ mice]) (Fig 4A–4C). Quantitative analysis demonstrated that the number

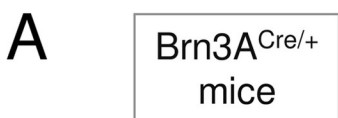

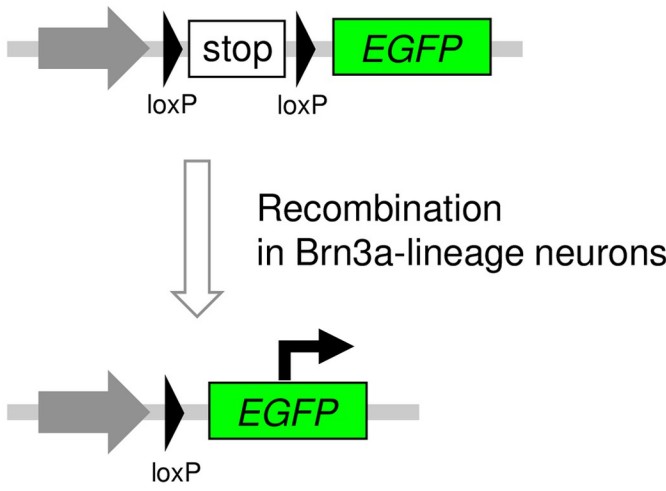

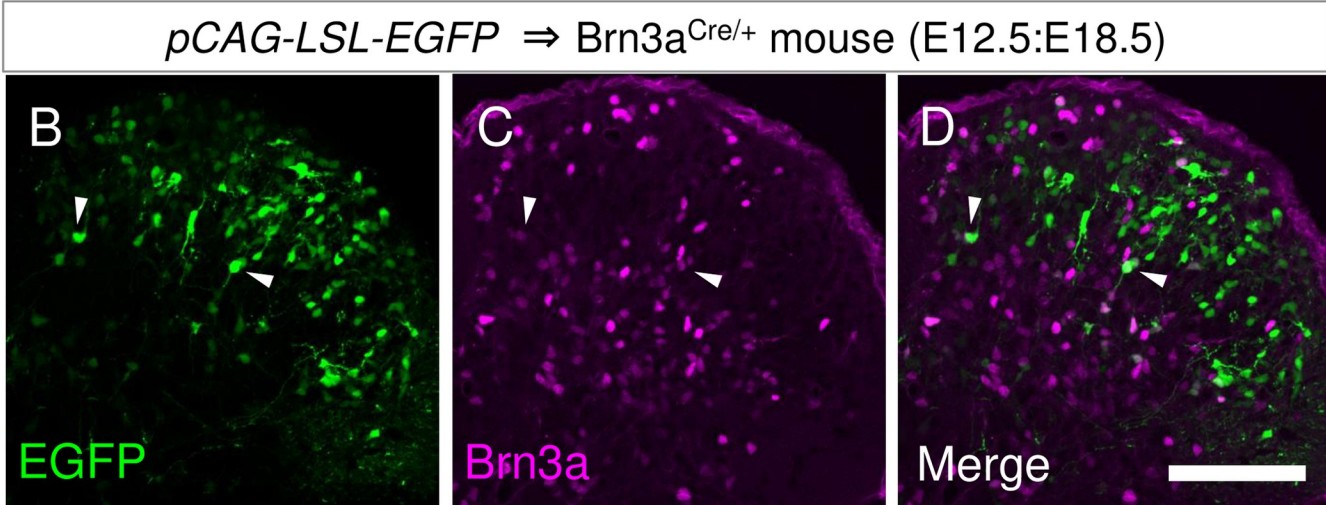

**Fig 2. Analysis of Brn3a-lineage neurons in the spinal dorsal horn.** (A) Introduction of a *pCAG-LSL-EGFP* construct into the spinal dorsal horn neurons of Brn3a[Cre/+] mice allows permanent labeling of Brn3a-lineage neurons by EGFP. (B-D) *pCAG-LSL-EGFP* was introduced into spinal dorsal horn neurons of Brn3a[Cre/+] mice at E12.5 by *in utero* electroporation, and the spinal cord of the mice was dissected out at E18.5. Transverse sections of the spinal cord were immunostained with anti-Brn3a antibody. EGFP fluorescence (green; B, D) together with Brn3a immunostaining (magenta; C, D) on the right dorsal horn are shown. Scale, 100 μm.

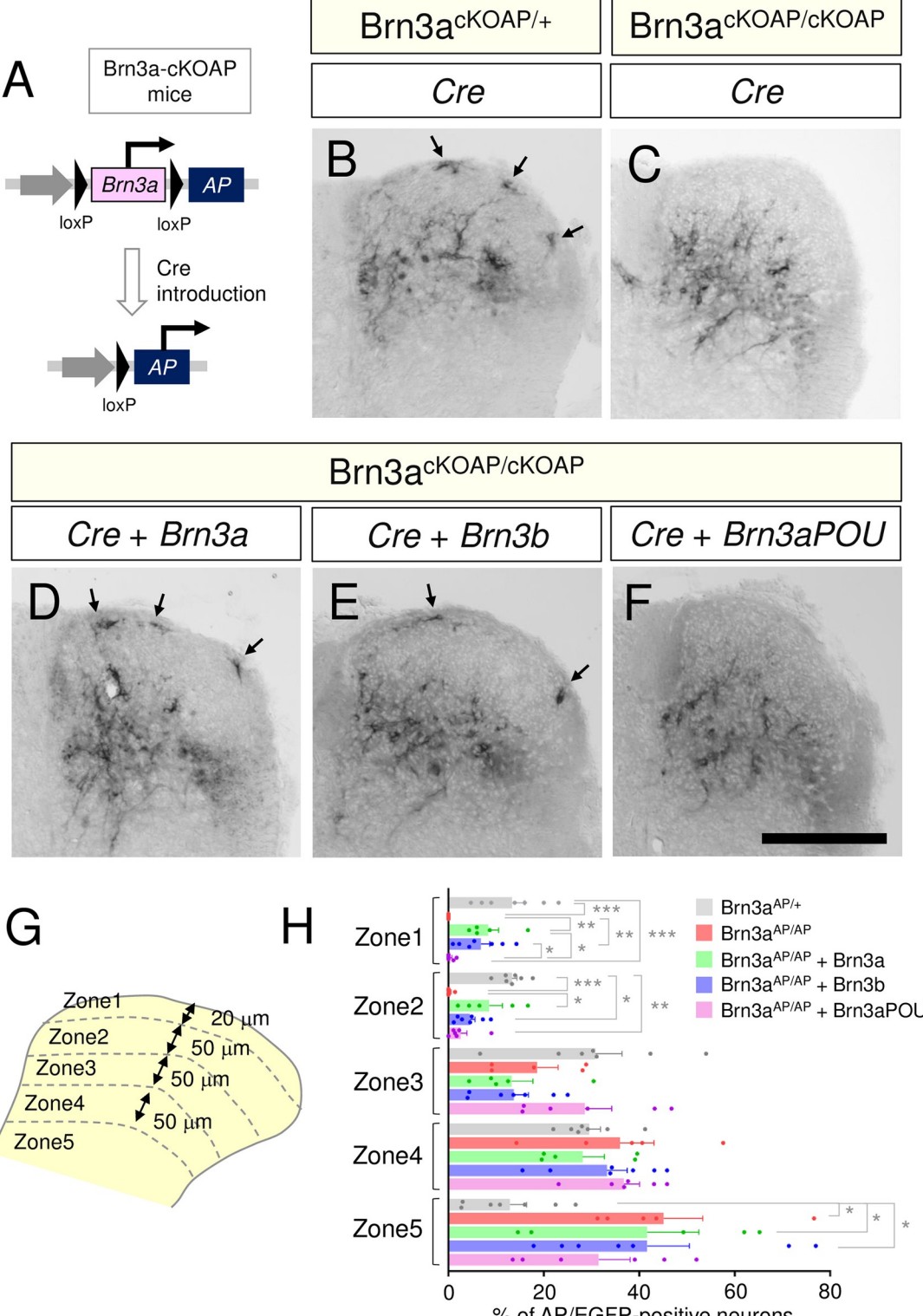

**Fig 3. Effect of Brn3a deficiency on the soma localization of Brn3a-persistent spinal dorsal horn neurons.** (A) Analysis of Brn3a-KO neurons using Brn3a-cKOAP mice. Following Cre-mediated deletion of the Brn3a coding region, AP was subsequently expressed specifically in Brn3a-persistent neurons. (B-F) Cre expression vector was introduced into the spinal dorsal horn neurons of Brn3a^cKOAP/+ mice at E11.8 via *in utero* electroporation, and the spinal cord samples were dissected at E18.5 (B). The Cre expression vector without (C) or with Brn3a (D), Brn3b (E), or Brn3aPOU (F) expression vectors was

introduced into the spinal dorsal horn neurons of Brn3a$^{cKOAP/cKOAP}$ mice at E11.8, and the spinal cord samples were dissected at E18.5. AP assay was performed on the transverse sections of the samples. Images of AP assay on the right spinal dorsal horn are shown. Arrows indicate AP-positive cells in the marginal zone. Scale, 200 μm. (G) Five zones were defined in the spinal dorsal horn according to the distance from the dorsal surface. Schematic of the right spinal dorsal horn is shown. (H) Percentage of AP- and EGFP-positive neurons in each zone was analyzed in Brn3a$^{AP/+}$ (Brn3a$^{cKOAP/+}$ mice with Cre expression [$n$ = 7 mice]), Brn3a$^{AP/AP}$ (Brn3a$^{cKOAP/cKOAP}$ mice with Cre expression [$n$ = 5 mice]), Brn3a$^{AP/AP}$ + Brn3a (Brn3a$^{cKOAP/cKOAP}$ mice with Cre and Brn3a expression [$n$ = 5 mice]), Brn3a$^{AP/AP}$ + Brn3b (Brn3a$^{cKOAP/cKOAP}$ mice with Cre and Brn3b expression [$n$ = 7 mice]), and Brn3a$^{AP/AP}$ + Brn3a-POU (Brn3a$^{cKOAP/cKOAP}$ mice with Cre and Brn3a-POU expression [$n$ = 6 mice]). Significant differences were assessed using the Kruskal–Wallis test, followed by the Benjamini, Krieger, and Yekutieli method. [Zone 1] Brn3a$^{AP/+}$ vs Brn3a$^{AP/AP}$, $q$ = 0.0007; Brn3a$^{AP/+}$ vs Brn3a$^{AP/AP}$ + Brn3a-POU, $q$ = 0.0009; Brn3a$^{AP/AP}$ vs Brn3a$^{AP/AP}$ + Brn3a, $q$ = 0.0088; Brn3a$^{AP/AP}$ vs Brn3a$^{AP/AP}$ + Brn3b, $q$ = 0.0094; Brn3a$^{AP/AP}$ + Brn3a vs Brn3a$^{AP/AP}$ + Brn3a-POU, $q$ = 0.0123; and Brn3a$^{AP/AP}$ + Brn3b vs Brn3a$^{AP/AP}$ + Brn3a-POU, $q$ = 0.015. [Zone 2] Brn3a$^{AP/+}$ vs Brn3a$^{AP/AP}$, $q$ = 0.0003; Brn3a$^{AP/+}$ vs Brn3a$^{AP/AP}$ + Brn3b, $q$ = 0.0447; Brn3a$^{AP/+}$ vs Brn3a$^{AP/AP}$ + Brn3a-POU, $q$ = 0.0074; and Brn3a$^{AP/AP}$ vs Brn3a$^{AP/AP}$ + Brn3a, $q$ = 0.0109. [Zone 5] Brn3a$^{AP/+}$ vs Brn3a$^{AP/AP}$, $q$ = 0.0264; Brn3a$^{AP/+}$ vs Brn3a$^{AP/AP}$ + Brn3a, $q$ = 0.0488; and Brn3a$^{AP/+}$ vs Brn3a$^{AP/AP}$ + Brn3b, $q$ = 0.0264. *$q$ < 0.05, **$q$ < 0.01, ***$q$ < 0.001. Data are presented as the mean ± SEM.

of Zfhx3-positive neurons in the marginal lamina in Brn3a$^{-/-}$ mice was significantly lower than that in Brn3a$^{+/+}$ mice (Brn3a$^{+/+}$ mice = 34.4 ± 1.8 cells per section [$n$ = 6 mice], Brn3a$^{-/-}$ mice = 28.6 ± 1.0 cells per section [$n$ = 8 mice], $p$ = 0.008, Mann–Whitney $U$-test) (Fig 4D and 4E). These results suggest that Brn3a plays critical roles in the localization of Brn3a-persistent neurons in the marginal laminae, and that Brn3b is functionally equivalent to Brn3a in the developmental process.

The C-terminal POU-homeobox domain of Brn3a is crucial for DNA-binding (S6A Fig) [6]. However, the role of its N-terminal region remains unknown. Therefore, we constructed a truncated mutant of Brn3a lacking the N-terminal region (Brn3a-POU) and examined its ability to rescue the phenotype of Brn3a deficiency. This mutant transactivated TrkA, a downstream target of Brn3a [34], as efficiently as the full-length Brn3a *in vitro* (S6B Fig). However, its expression did not rescue the lack of Brn3a$^{AP/AP}$ neurons in zones 1 and 2 (S6C–S6H Fig, Fig 3F and 3H, S1 File). This result raises the possibility that the N-terminal region of Brn3a is required for the marginal localization of spinal dorsal horn neurons.

## Loss of Brn3a did not affect overall axonal extension of Brn3a-persistent neurons

Next, we focused on the effect of Brn3a deficiency on the axonal extension of Brn3a-persistent neurons. Since the axonal morphology and synaptic targets of Brn3a-persistent spinal dorsal horn neurons remain unknown, we first analyzed them by labeling with two fluorescent proteins, tdTomato and synaptophysin-fused EGFP (SypEGFP), to tag the cytoplasm and presynaptic terminals of Brn3a-persistent neurons, respectively. AAV carrying a Cre-dependent expression cassette of these genes [27] was unilaterally injected into the lumbar spinal dorsal horn of Brn3a-Cre knock-in mice [13] (S7A Fig), and the fluorescence of tdTomato and EGFP was analyzed throughout the spinal cord and brain. The axonal tract of Brn3a-persistent neurons was mainly observed in five different regions in the spinal cord: (i) dorsal funiculus (DF) on the ipsilateral side, (ii) gray matter on the ipsilateral side, (iii) dorsolateral funiculus (DLF) on the ipsilateral side, (iv) ventral funiculus (VF) and ventrolateral funiculus (VLF) on the ipsilateral side, and (v) VF and VLF on the contralateral side (S7D Fig). Many axons on the contralateral side were localized around the VF at the rostrocaudal level close to the injection site (+0.5 cm, S7D Fig), whereas those localized in the VLF were detected at more rostral levels (+1.0 cm−+2.0 cm, S7D Fig). In contrast, axons derived from Brn3a-persistent neurons were rarely found in the ipsilateral VF. Additionally, axons were detected at the spinal level caudal to the injection site (-0.5 or -1.0 cm, S7D Fig), indicating that Brn3a-persistent neurons extend axons toward the caudal spinal cords. Some of the rostrally extending axons derived from

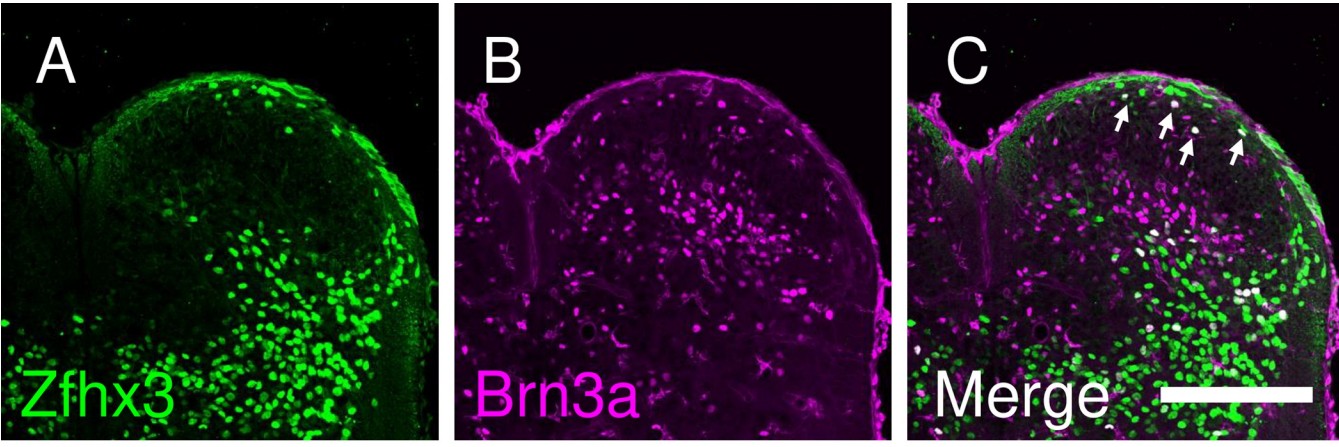

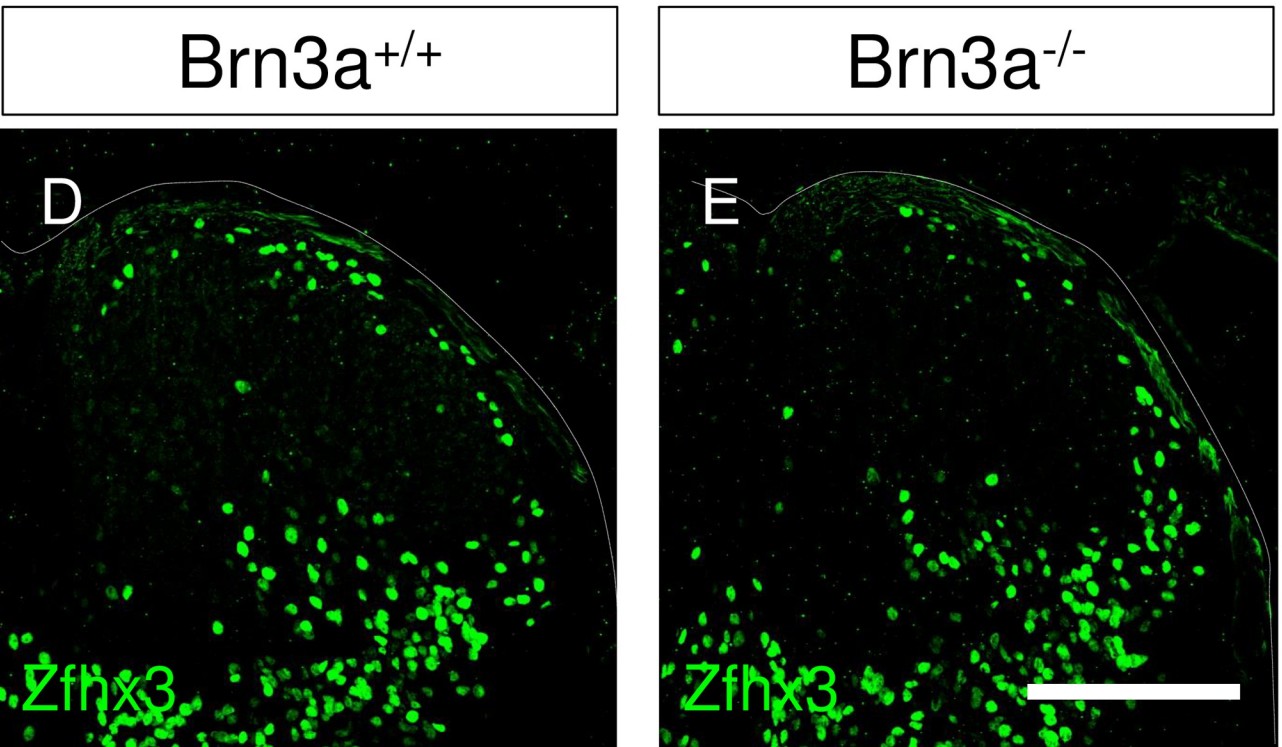

**Fig 4. Reduction of the marginal Brn3a-persistent spinal dorsal horn neurons analyzed by the expression of Zfhx3.** (A-C) Immunostaining of Zfhx3 (green; A, C) and Brn3a (magenta; B, C) was performed on the transverse section of the lumbar spinal cord of C57BL/6J mice at E18.5. Fluorescence images on the right spinal dorsal horn are shown. Arrows indicate double-positive neurons in the marginal area. Scale, 200 μm. (D, E) Immunostaining of Zfhx3 was performed on the transverse section of the lumbar spinal cord of Brn3a$^{+/+}$ (D) and Brn3a$^{-/-}$ mice (E) at E18.5. Dotted lines indicate the dorsal surface of the spinal cord. Scale, 200 μm.

Brn3a-persistent neurons appeared to pass beyond the cervical level (S7D Fig top) and eventually formed synaptic connections with several nuclei in the medulla and pons, such as the gracile nucleus (GN), lateral parabrachial nucleus (LPb), and nucleus of the solitary tract (NTS) (S7E–S7M and S8 Figs).

Based on these results, we re-examined Brn3a-cKOAP mice expressing Cre (Fig 3B and 3C) to analyze axonal distribution in the white matter. Since AP signals derived from axons were much weaker than those derived from cell bodies, we performed a longer enzymatic reaction with AP (Fig 5, S9 Fig). The AP signal in the white matter of Brn3a$^{cKOAP/+}$ mice treated with Cre was detected in the DF and LF on the ipsilateral side as well as in the VF on the contralateral side (Fig 5A–5D). This localization pattern at E18.5 is in accordance with that of the axonal tract of Brn3a-persistent neurons in adults (S7D Fig). In Brn3a$^{cKOAP/cKOAP}$ mice (Fig 5E–5H), the AP signal in the white matter was comparable to that in Brn3a$^{cKOAP/+}$ mice. These results suggest that loss of Brn3a does not affect overall axonal extension of Brn3a-persistent neurons toward the ipsilateral DF, LF, and contralateral VF.

## Brn3a overexpression directs the soma localization in a manner similar to Brn3a-persistent neurons

As described in Fig 2, Brn3a was downregulated in most Brn3a-lineage neurons at E12.5 (Brn3a-transient neurons), and only 13.5 ± 2.5% of the Brn3a-lineage neurons continued to express Brn3a at E18.5 (Brn3a-persistent neurons). If Brn3a plays a critical role in the cell fate specification of Brn3a-persistent neurons, prolonged Brn3a expression in Brn3a-transient neurons is expected to confer the characteristics of Brn3a-persistent neurons, such as their specific localization pattern. To examine this possibility, we introduced a Cre-dependent EGFP expression vector (*pCAG-LSL-EGFP*; Fig 2) with or without the Brn3a-expression vector into Brn3a-lineage neurons at E12.5 (Fig 6) and analyzed the localization of EGFP-positive cells at E18.5. In Brn3a$^{Cre/+}$ mice expressing *pCAG-LSL-EGFP* (control), EGFP-positive neurons were mainly localized around the shallow dorsal horn at E18.5 (Fig 6A), as described in Fig 2. In contrast, neurons overexpressing with Brn3a (Brn3a overexpression) were localized in a broader region of the spinal dorsal horn (Fig 6B), with a pattern resembling the distribution of Brn3a-persistent neurons at this stage (S5C Fig). We quantitatively analyzed the distribution of EGFP-positive neurons according to their distance from the surface (Fig 6C and 6D, S2 File), as described above (Fig 3G). The percentage of Brn3a-overexpressing neurons in zone 2 was significantly lower than that of control neurons, whereas the percentage of Brn3a-overexpressing neurons in zone 1, 4, and 5 was significantly higher than that of control neurons (zone 1: control = 13.8 ± 1.6% [*n* = 5 mice] and Brn3a = 17.2 ± 0.8% [*n* = 5 mice], *p* = 0.0397; zone 2: control = 58.1 ± 1.0% [*n* = 5 mice] and Brn3a = 24.0 ± 1.8% [*n* = 5 mice], *p* = 0.0079; zone 4: control = 7.7 ± 0.4% [*n* = 5 mice] and Brn3a = 26.3 ± 2.4% [*n* = 5 mice], *p* = 0.0079; zone 5: control = 0.8 ± 0.1% [*n* = 5 mice] and Brn3a = 15.0 ± 3.0% [*n* = 5 mice], *p* = 0.0079, Mann–Whitney *U*-test) (Fig 6D). These results suggest that prolonged Brn3a expression directs Brn3a-transient neurons to localize in a manner similar to Brn3a-persistent neurons.

## Brn3a overexpression directs the axonal extension in a manner similar to Brn3a-persistent neurons

Many more axons in the VF and VLF appeared to have been derived from Brn3a-overexpressing than from control neurons (arrows in Fig 6B), raising the possibility that prolonged Brn3a expression in Brn3a-transient neurons directs their axonal extension in a manner similar to that of Brn3a-persistent neurons (S8 Fig). Thus, we quantitatively analyzed the results by normalizing the number of labeled axons to the number of labeled cell bodies to calculate the axonal extension index, as the efficiency of gene transfer via *in utero* electroporation varies across samples. The index of the ipsilateral DF in Brn3a-overexpressing neurons was not significantly different from that in control ones (control = 0.19 ± 0.04 [*n* = 5 mice] and Brn3a = 0.20 ± 0.03 [*n* = 5 mice], *p* = 0.6905, Mann–Whitney *U*-test) (Fig 7A, S3 File). In contrast, the index of the

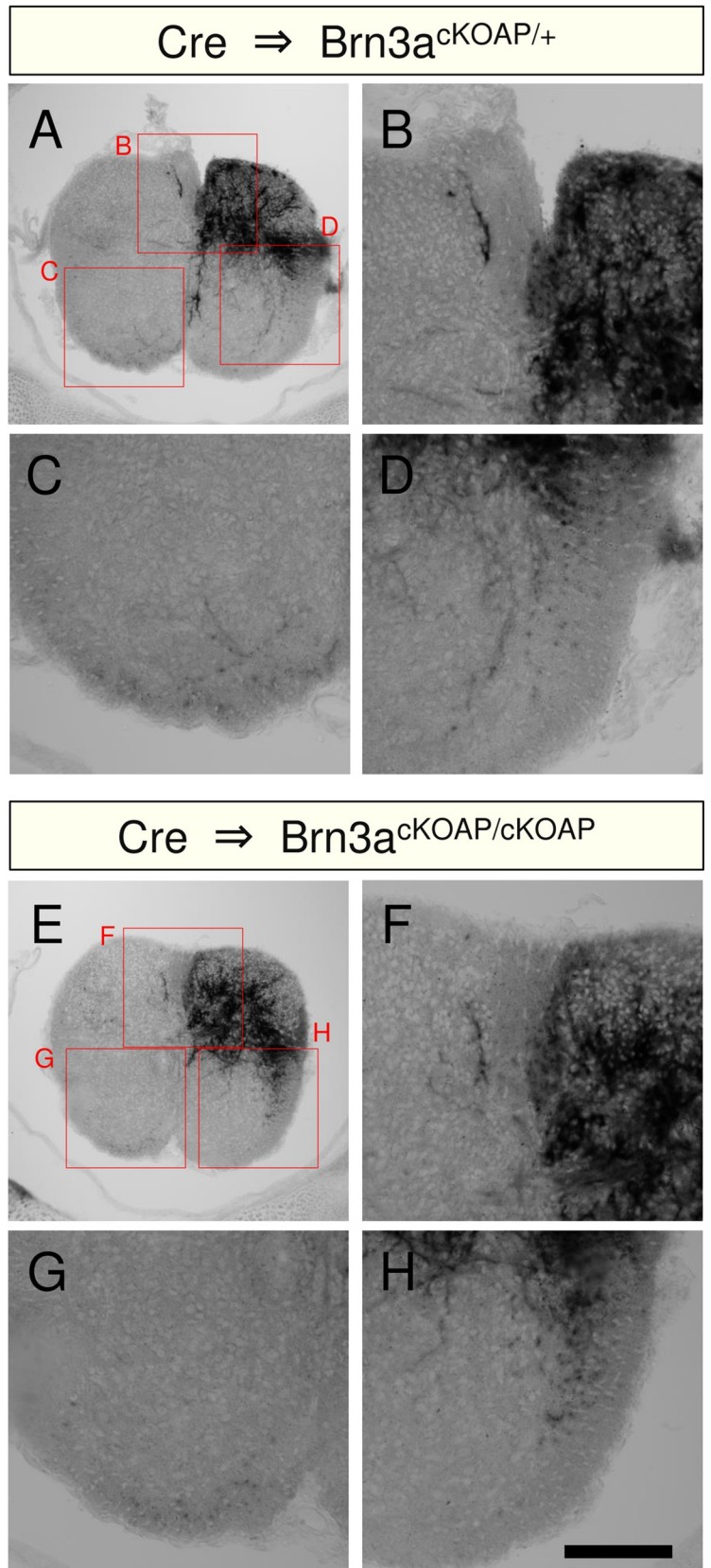

**Fig 5. Effect of Brn3a deficiency on the axonal extension of Brn3a-persistent neurons in the white matter.** Cre expression vector was introduced into the spinal cord of Brn3a$^{cKOAP/+}$ (A-D) and Brn3a$^{cKOAP/cKOAP}$ (E-H) mice, and AP assay was performed on the transverse section of the samples. Low magnification images of AP staining are shown in A and E. High-magnification views (marked in A and E) of AP staining around the DF (B, F), ventral funiculus on the contralateral side (C, G), and VLF on the ipsilateral side (D, H) are shown. Scale, 100 μm.

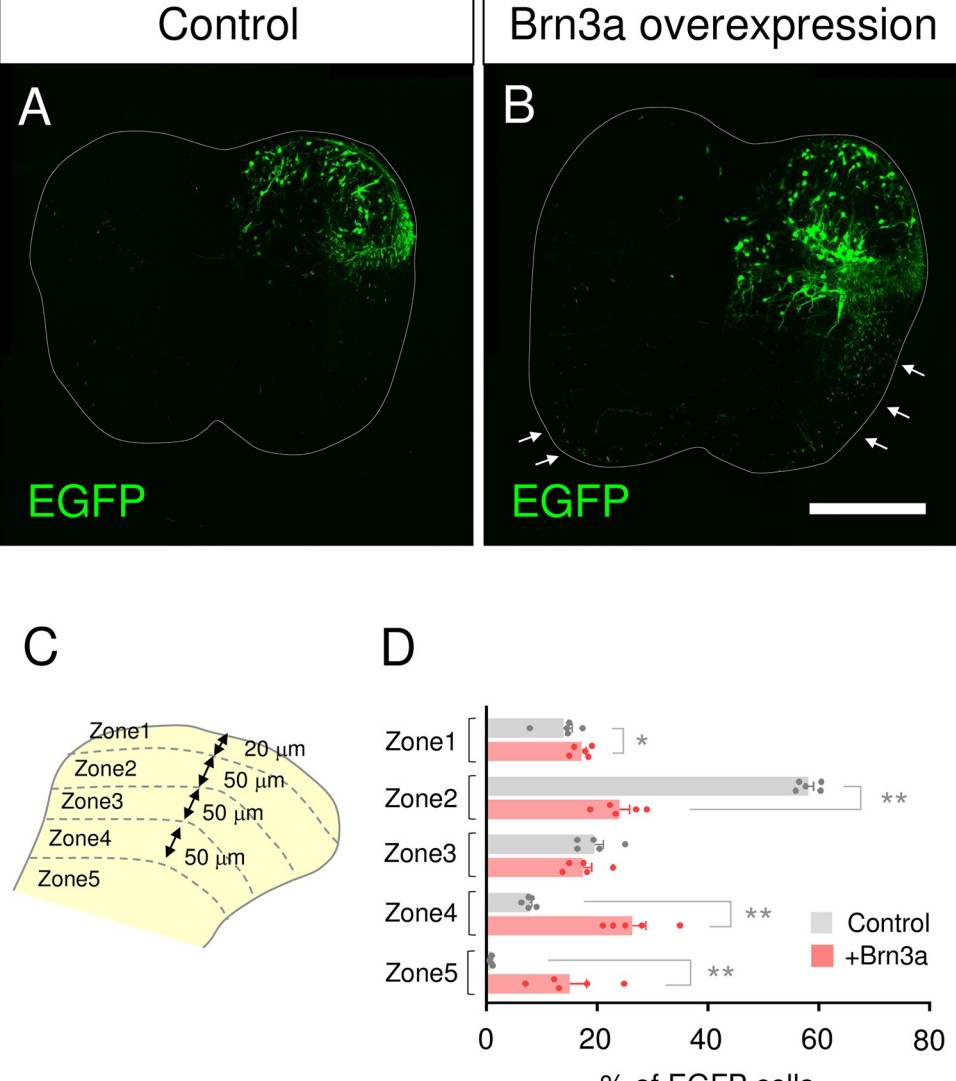

**Fig 6. Effect of Brn3a overexpression on the localization of Brn3a-lineage neurons.** (A, B) *pCAG-LSL-EGFP* without (A, control) or with *pCAG-Brn3a* (B, Brn3a overexpression) was unilaterally introduced into the spinal dorsal horn neurons of Brn3a$^{Cre/+}$ mice at E12.5 via *in utero* electroporation, and the transverse sections of the spinal cord at E18.5 were prepared to analyze the localization of EGFP-positive neurons. Arrows indicate EGFP-positive fibers derived from Brn3a-overexpressing neurons in the VF and VLF. Scale, 200 μm. (C) Five zones were defined in the spinal dorsal horn as shown in Fig 3G. (D) Percentage of EGFP-positive neurons localized in each zone of the spinal cord in control (*n* = 5) and Brn3a-overexpressing (*n* = 5) mice is shown. Significant differences were assessed using the Mann–Whitney *U*-test. Zone 1, *p* = 0.0397; Zone 2, *p* = 0.0079; Zone 3, *p* = 0.3889; Zone 4, *p* = 0.0079; Zone 5, *p* = 0.0079. \**p* < 0.05, \*\**p* < 0.01. Data are the mean ± SEM.

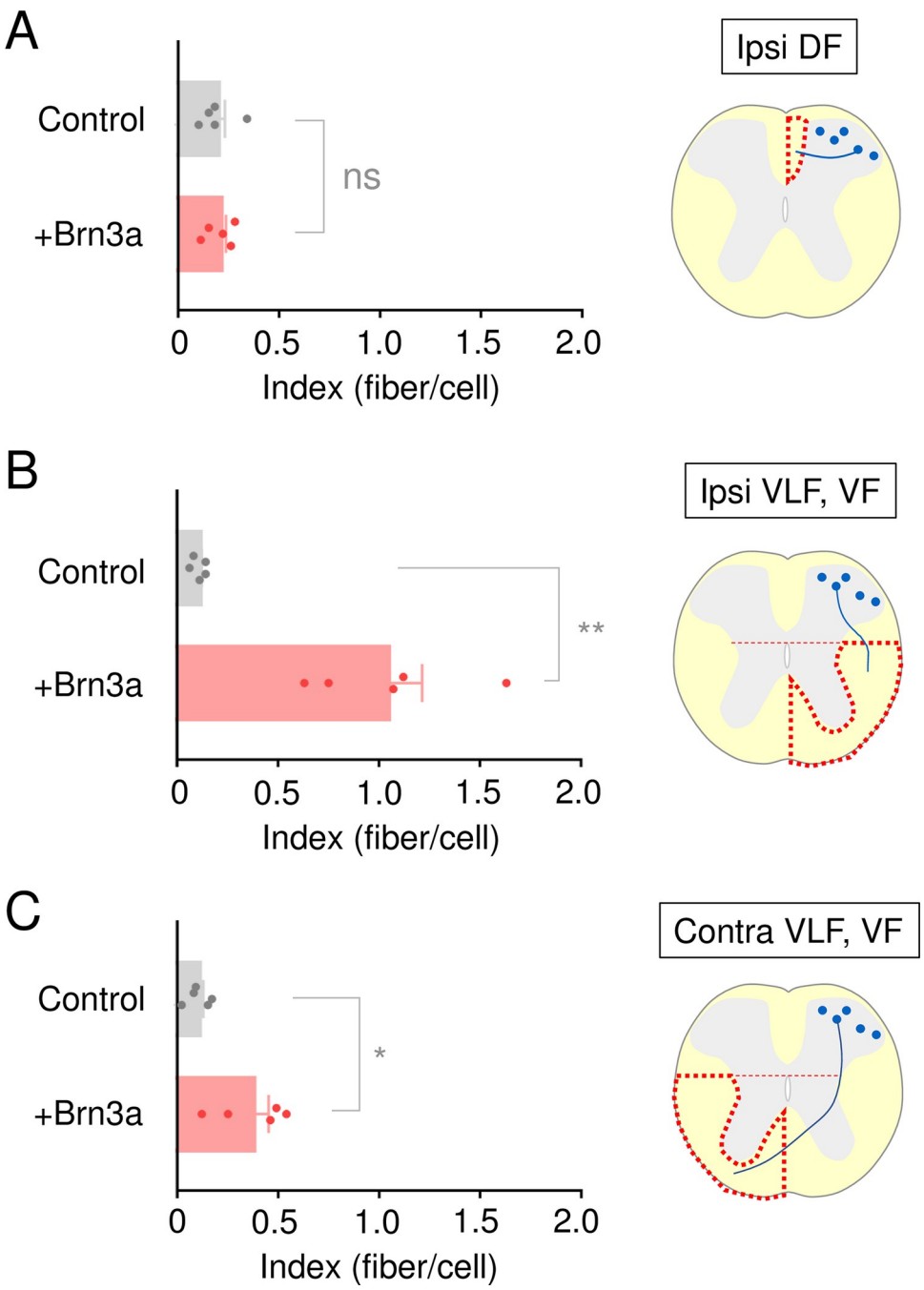

**Fig 7. Effect of Brn3a overexpression on the axonal extension of Brn3a-lineage neurons.** *pCAG-LSL-EGFP* with or without *pCAG-Brn3a* was introduced into the spinal dorsal horn neurons of Brn3a[Cre/+] mice, as shown in Fig 6. The number of EGFP-positive axons normalized by that of EGFP-positive cell bodies in each section was defined as the index. The index of control (*n* = 5) and Brn3a-overexpressing (*n* = 5) mice in the DF on the ipsilateral side (A) as well as in the VF and VLF on the ipsilateral (B) and contralateral (C) sides is shown. Significant differences were assessed using the Mann–Whitney *U*-test. (A) *p* = 0.6905. (B) *p* = 0.0079. (C) *p* = 0.0317. "ns" = non-significant. $^*p < 0.05$, $^{**}p < 0.01$. Data are the mean ± SEM.

VLF and VF in Brn3a-overexpressing neurons was significantly higher than that in control ones (Ipsi control = 0.11 ± 0.02 [$n$ = 5 mice], Ipsi Brn3a = 1.04 ± 0.17 [$n$ = 5 mice], $p$ = 0.0079; Contra control = 0.10 ± 0.03 [$n$ = 5 mice], Contra Brn3a = 0.37 ± 0.08 [$n$ = 5 mice], $p$ = 0.0317, Mann–Whitney $U$-test) (Fig 7B and 7C, S3 File). Further, we analyzed the axonal extension of Brn3a-overexpressing neurons in the LF and VLF in the rostrocaudal direction and found that most of them did not extend their axons long distances from the spinal segment where their cell bodies resided (S10 Fig). These results suggest that prolonged Brn3a expression in Brn3a-transient neurons directs axonal extension toward the VF and VLF, where the axons of Brn3a-persistent neurons reside.

## Discussion

In the present study, we analyzed the role of the POU family transcription factor Brn3a in the development of spinal dorsal horn neurons. Loss of Brn3a disturbed the cell body localization of Brn3a-persistent neurons in the dorsal horn but did not affect their overall axonal extension. In contrast, prolonged Brn3a expression in Brn3a-lineage neurons directed cell body localization and axonal extension in a manner similar to that in Brn3a-persistent neurons. These results suggest that Brn3a is involved in the development of Brn3a-persistent spinal dorsal horn neurons.

### Technical consideration of Brn3a-KO in the present study

For spinal dorsal horn-specific KO of Brn3a, we first utilized Lbx1-Cre mice [14], which are widely used as spinal dorsal horn-specific driver lines. However, nonspecific recombination occurred in a population of DRG neurons in Lbx1$^{Cre/+}$; Brn3a$^{cKOAP/+}$ mice (S11 Fig). Brn3a-KO in the DRG severely suppresses the entry of primary sensory afferents into the spinal cord [35], which is required for the proper migration of superficial spinal dorsal horn neurons [5]. Therefore, we reasoned that Lbx1$^{Cre/+}$;Brn3a$^{cKOAP/+}$ mice were not suitable for our purpose, as we could not distinguish between the cell-autonomous and non-cell-autonomous effects of Brn3a-KO in this strain. Thus, we performed gene transfer of Cre into Brn3a-cKOAP mice via *in utero* electroporation, which enabled spinal dorsal horn-specific Brn3a-KO. Owing to the technical difficulties in gene transfer via *in utero* electroporation before E11.5, we performed this experiment at E11.8. Birthdate analysis of Brn3a-persistent neurons in the spinal dorsal horn demonstrated that approximately half of these neurons had exited the final cell cycle after E11.5 (S12 Fig). This result suggests that the phenotypes of Brn3a-KO mice observed in the present study can be applied to later-born populations of Brn3a-persistent neurons. Although we cannot exclude the possibility that the role of Brn3a in the early-born population is different from that in the late-born population, we believe that the reduction in the number of neurons in the marginal layer is a common defect in both early- and late-born Brn3a-persistent neurons, as this phenotype was also confirmed in Brn3a$^{-/-}$ mice (Fig 4). In the future, spinal dorsal horn-specific Cre driver lines would be more suitable for obtaining the complete picture of Brn3a-persistent neuronal development.

### Cell fate specification of Brn3a-transient and persistent neurons

Brn3a is expressed in almost all the spinal progenitors giving rise to excitatory dorsal horn neurons, including dI1, dI2, dI3, dI5, and dIL$_B$ [2]. The lineage analysis in the present study (Fig 2) demonstrated that most Brn3a-positive progenitors ceased to express Brn3a at E18.5 (Brn3a-transient neurons) and only a limited population of them persisted to express Brn3a (Brn3a-persistent neurons). Previous single cell RNA-seq studies demonstrated that Brn3a was expressed in 4 out of 15–16 subclasses of excitatory dorsal horn neurons in the adult [36, 37], which correspond to Brn3a-persistent neurons. Molecular mechanisms of how Brn3a-transient neurons downregulate Brn3a expression and how Brn3a-persistent neurons maintain its

expression at later stages remain unknown. It is possible that these two populations are derived from different progenitors, and transcription factors specifically expressed in the progenitors of Brn3a-transient neurons may suppress Brn3a expression to inhibit cell fate of Brn3a-persistent neurons. Alternatively, diversification of these two populations may be determined by a temporal mechanism, in which stage-specific transcription factors in the progenitors confer neuronal cell fate [38]. However, this possibility is less likely because Brn3a-persistent neurons include both early- and late-born populations (S12 Fig). Further investigation of these two populations would be needed to clarify the mechanism of their cell fate specification.

## Role of Brn3a in the soma localization of Brn3a-persistent neurons

Brn3a-KO (Brn3a$^{AP/AP}$) neurons were completely absent in the marginal zones, and those in the deeper zones were localized more ventrally than Brn3a-hetero (Brn3a$^{AP/+}$) neurons (Fig 3B, 3C, and 3H). This result raises the possibility that Brn3a controls the migration of Brn3a-persistent neurons from the ventricular zone to the shallow dorsal horn. Meanwhile, Brn3a-overexpressing neurons were localized more in the marginal and deeper zones than control neurons (Fig 6), as if the Brn3a-overexpressing neurons were repelled by the "Brn3a-void region." Given the dynamic changes in Brn3a expression in developing Brn3a-lineage neurons (Figs 1 and 2), Brn3a function may differ at each embryonic stage, which may better explain the phenotypic differences between Brn3a deficiency and overexpression: Brn3a at early stages (~E12.5) is required for a more general function in all early post-mitotic excitatory neurons, including both Brn3a-transient and -persistent neurons, while its role at later stages is more specific to Brn3a-persistent neurons. However, we cannot rule out the possibility that the absence of Brn3a$^{AP/AP}$ neurons in the marginal zones was caused by selective neuronal cell death. Analysis of marker molecules specific to marginal Brn3a-persistent neurons in Brn3a-KO mice would help clarify this possibility in the future.

The POU domain located in the C-terminus of the POU family transcription factors exhibits its high amino acid sequence similarity between members, and plays a pivotal role for the transactivation of downstream targets [6]. The ability of Brn3b to rescue the phenotype of Brn3a deficiency (Fig 3E and 3H) indicates that Brn3b is functionally equivalent to Brn3a in the spinal cord development. In accordance with our results, knock-in of Brn3a in the Brn3b locus rescued the phenotype of Brn3b deficiency in the retinal development [39]. On the other hand, the POU-domain of Brn3a failed to rescue the phenotype of Brn3a deficiency in Brn3a-persistent neurons (Fig 3F and 3H). This result is in contrast to the previous study in which the POU-domain of Brn3b or Brn3c induced differentiation of retinal ganglion cells as efficiently as the full-length [40]. This suggests that the N-terminal region of Brn3a, whose function has been previously under-appreciated, plays indispensable roles for some aspects of neuronal development. Indeed, in contrast to the downstream targets of Brn3a such as TrkA (S6 Fig), neurofilament [41], and NGFI-A [42], the POU-domain of Brn3a could not transactivate those such as Bcl-2 [43], Bcl-x [44], and Hsp27 [45]. Regarding the example of Bcl-2 shown above [43], overexpression of the N-terminal region of Brn3a inhibited transactivation of the Bcl-2 promoter by full-length Brn3a, raising the possibility that the N-terminal region interacts with transcriptional co-factors essential for this process. It is conceivable that the development of Brn3a-persistent spinal dorsal horn neurons requires the downstream targets of Brn3a whose expression depends on its N-terminal region.

## Axonal projection pattern of Brn3a-persistent spinal dorsal horn neurons

The lack of knowledge regarding the synaptic connectivity and axonal extension patterns of Brn3a-persistent spinal dorsal horn neurons has hampered our understanding of their

neuronal identity and role in sensory transmission. The present study, using anterograde labeling of Brn3a-persistent neurons, elucidated their innervation of a wide array of brain regions, such as the GN, LPb, and NTS (S7 and S8 Figs). In addition, Brn3a-persistent neurons appeared to extend their axons toward several distant regions inside the spinal cord. They innervate a small region in the deep dorsal horn (+0.5 cm, +1.0 cm; S7D Fig) as well as send descending axons at least 0.5 cm farther from the injection site (-0.5 cm; S7D Fig). Moreover, the number of ascending axons in the DF, DLF, and VLF around the rostral spinal cord (e.g., +2.0 cm; S7D Fig) appeared to be much lower than that near the injection site (e.g., +0.5 cm; S7D Fig), suggesting that some of the ascending axons exited from the white matter to innervate the distant spinal cord. Corroborating our observations, both supraspinal and propriospinal neurons appear to exhibit similar localization patterns as Brn3a-persistent neurons: both spino-parabrachial projection [46] and long-range propriospinal [47] neurons reside in the marginal and deeper laminae, whereas post-synaptic dorsal column (PSDC) projection neurons innervating the GN are localized exclusively to the deeper laminae [48].

Only a small population of Brn3a-persistent neurons expressed Zfhx3 (S5 Fig), which was recently shown to be a marker for long-range spinal dorsal horn neurons [33]. Thus, one can postulate that among Brn3a-persistent neurons Zfhx3-positive and -negative populations constitute long- and short-range (local-circuit) neurons, respectively. However, considering the increased axonal extension of Brn3a-overexpressing neurons toward the VF and VLF (Fig 7), a majority of Brn3a-persistent neurons likely have the characteristics of long-range spinal dorsal horn neurons. The spinal dorsal horn contains a variety of long-range propriospinal neurons that innervate other spinal segments at different distances [47], and they may be candidates of Zfhx3-negative Brn3a-persistent neurons. Further investigation of Brn3a-persistent neurons is required to clarify their identities.

## Role of Brn3a in the axonal extension of Brn3a-persistent neurons

Loss of Brn3a did not affect the overall axonal extension of Brn3a-persistent neurons toward the DF, LF, or VF (Fig 5), although quantitative analyses could not be performed owing to technical limitations. In contrast, Brn3a overexpression significantly increased axonal extension toward the VF and VLF (Fig 7), where the axonal tract of Brn3a-persistent long-range neurons lies (S7 Fig). This phenotype is consistent with the cell body localization pattern of Brn3a-overexpressing neurons (Fig 6). These results suggest that Brn3a is not essential but sufficient for axonal extension toward the VF and VLF. The lack of an obvious phenotype in Brn3a-KO mice may be due to the functional redundancy of transcription factors involved in the developmental process. The molecular mechanism underlying the axon guidance of Brn3a-persistent spinal dorsal horn neurons was not addressed in the present study. However, the preferential axonal extension of Brn3a-overexpressing neurons toward the VF and VLF but not the DF suggests that Brn3a transactivates the downstream targets required to guide axons toward the ventral cord, such as receptors for axonal attractants or repellants. Netrin1 and Sonic hedgehog expressed in the floor plate function as attractive axonal cues toward the ventral cord, whereas Bmp expressed in the roof plate functions as a repellent for the dorsal cord [49]. Axons in the VF and VLF derived from Brn3a-overexpressing neurons did not extend along the rostrocaudal axis (S10 Fig), indicating that Brn3a overexpression does not completely confer the characteristics of long-range spinal dorsal horn neurons. This may be due to the lack of induction of guidance molecules required for this process, such as Wnt signaling components [50]. Alternatively, this may be attributable to differences in the developmental timing critical for axonal extension along the rostrocaudal axis. In our previous study, we demonstrated that the birthdate of long-range neurons is much earlier than that of local

circuit neurons [4]. It is possible that the axonal outgrowth of these neurons requires a certain environment of spinal cord at earlier stages.

## Future perspectives

The phenotypes of Brn3a deficiency and overexpression indicate the involvement of Brn3a in the development of Brn3a-persistent spinal dorsal horn neurons, which constitute subsets of long-range neurons. Transcriptomic analyses in several regions of the nervous system demonstrated that the expression of many genes is under the control of Brn3a [21, 51–54]. However, these Brn3a target genes differ across regions, suggesting that Brn3a controls the expression of different subsets of genes in specific regions of the nervous system, possibly via interactions with region-specific co-factors. Thus, identification of the downstream targets of Brn3a in spinal dorsal horn neurons is crucial for further understanding the developmental program of Brn3a-persistent spinal dorsal horn neurons. We previously showed that a subset of Brn3a-persistent spinal dorsal horn neurons located in the marginal region is over-represented among visceral pain-responsive neurons [9]. In the present study, we found that loss of Brn3a disrupted the distribution of Brn3a-persistent spinal dorsal horn neurons (Fig 3). Therefore, elucidating the role of Brn3a in the development of this subset of neurons will further our understanding of the transmission of visceral pain in the spinal cord.

## Supporting information

**S1 Fig. Generation of Brn3b-CreERT-Neo mice.** (A) Schematic representation of the wild-type and mutant Brn3b allele (Brn3b$^{CreERT-Neo}$). The first exon of the Brn3b gene downstream of the translation initiation codon was replaced by a CreERT-FRT-Neo-ERT cassette to generate Brn3b$^{CreERT-Neo}$ allele. (B) Southern blot analysis of MfeI-, BmtI-, and KpnI-digested genomic DNA from Brn3b$^{CreERT-Neo/+}$ ES cells. The 5' probe identifies MfeI fragments of 13.1 kb (wild-type) and 17.6 kb (mutant). The 3' probe identifies BmtI fragments of 9.0 kb (wild-type) and 13.5 kb (mutant). The neo probe identifies a KpnI fragment of 13.5 kb (mutant). Arrows and arrowheads indicate fragments derived from wild-type and mutant alleles, respectively. (PDF)

**S2 Fig. Spinal dorsal horn-specific gene transfer by in utero electroporation.** *pCAG-mCherry* was unilaterally introduced into spinal dorsal horn neurons of Brn3a$^{Cre/+}$ mice at E12.5 by *in utero* electroporation, and the spinal cord of the mice was dissected at E18.5. Fluorescence of mCherry (magenta; A, B) and Hoechst 33342 (blue; B) on the transverse section of the sample is shown. White dotted lines indicate the outline of the spinal cord and DRG. Note that mCherry-positive cells are only found in the spinal cord but not DRG. Scale, 200 μm. (PDF)

**S3 Fig. Analysis of Brn3b-lineage neurons in the spinal dorsal horn.** Tamoxifen was intra-peritoneally injected into the pregnant Brn3b$^{CreERT/+}$;Ai9 mice from E10.5 to E14.5 to label Brn3b-lineage spinal dorsal horn neurons. Transverse sections of the spinal cord of the mice at E18.5 were immunostained with anti-Brn3b antibody. (A) Fluorescence of tdTomato in the spinal cord of the mice is shown. White dotted line indicates the outline of the spinal cord. (B-D) High magnification views (marked in A) of tdTomato fluorescence (magenta; B, D) together with Anti-Brn3b immunostaining (green; C, D) are shown. Arrowheads in D indicate double-positive neurons. The percentage of Brn3b-positive neurons among tdTomato-positive ones was 43.4 ± 1.5% (447 cells [$n$ = 4 mice]). Scale, 100 μm. (PDF)

**S4 Fig. Labeling of Brn3a-KO neurons by using Brn3a-cKOAP mice.** (A) *pCAG-Cre* together with *pCAG-nlsEGFP* were introduced into spinal dorsal horn neurons of Brn3a-cKOAP mice at E11.8 by *in utero* electroporation, and the spinal cord was dissected out at E18.5. AP assay was performed on the transverse section of the spinal cord to visualize the distribution of Brn3a-KO neurons. (B-D) AP signal (magenta; B, D) and nlsEGFP fluorescence (green; C, D) on the right spinal dorsal horn are shown. Arrows indicate double-positive neurons. Scale, 200 μm.
(PDF)

**S5 Fig. Expression of marker molecules in each zone of the spinal dorsal horn.** (A-E) Immunostaining of Zfhx3 (A), Brn3a (C), and CGRP (E), together with *in situ* hybridization of Pdyn (B) and CCK (D) were performed on the transverse section of the spinal cord of C57BL6J mice at E18.5. White solid lines indicate the boundary between gray and white matters. White dotted lines indicate the boundary of zones. (F) Schematic diagram of the expression pattern of marker molecules in each zone of the spinal dorsal horn at E18.5. Orange dotted line indicates the expression of Zfhx3 in the lateral edge of zone 2–4.
(PDF)

**S6 Fig. Deletion mutants of Brn3a.** (A) Schematic diagram of full-length Brn3a and its truncation mutants. The black and hatched boxes indicate the POU homeobox domain and FLAG-tag, respectively. Mouse Brn3a contains 421 amino acids. FLAG-Brn3a-ΔPOU lacks C-terminal half of the POU homeodomain. FLAG-Brn3a-POU contains POU homeodomain and five C-terminal amino acids. (B) Transcriptional activities of Brn3a and its truncation mutants. *pGL2B* or *pGL2B-Trk enhancer* together with indicated Brn3a constructs were transfected into COS7 cells, and the luciferase activity of the lysate measured 2 days after transfection. The activity of firefly luciferase was normalized by that of co-transfected *Renilla* luciferase. Fold luciferase activity relative to the control sample (*pGL2B* plus *pCAG*) is shown (pGL2B, 0.03 ± 0.00; pGL2B+Brn3a-Full, 0.15 ± 0.01; pGL2B-TrkA+Brn3a-Full, 6.82 ± 0.21; pGL2B-TrkA+Brn3aΔPOU, 1.21 ± 0.01; pGL2B-TrkA+Brn3aPOU, 5.25 ± 0.09). Horizontal bars indicate the median. (C-H) FLAG-Brn3a-POU overexpressed in the spinal dorsal horn. *nlsEGFP* together with *FLAG-Brn3a-POU* were introduced into spinal dorsal horn neurons of the mice at E12.5 by *in utero* electroporation. The spinal cord of the mice was dissected at E18.5, and transverse sections of the mice were immunostained with the anti-FLAG antibody. Fluorescence images of nlsEGFP (green; C, E) and anti-FLAG immunostaining (magenta; D, E) are shown. White dotted lines indicate the outline of the spinal cord. High magnification images (marked in C) of nlsEGFP (green; F, H) and anti-FLAG immunostaining (magenta; G, H) are shown. Scale, 100 μm.
(PDF)

**S7 Fig. Analysis of axonal tracts and terminals of Brn3a-persistent spinal dorsal horn neurons.** (A) AAV carrying FLEX-switched tdTomato and SypEGFP genes was injected into the lumbar spinal cord (~L5) of Brn3a$^{Cre/+}$ mice. This experiment enables labeling of Brn3a-persistent neurons and their presynaptic terminals by tdTomato and SypEGFP, respectively. (B-M) Three weeks after AAV injection, the spinal cord and whole brain of the mice were dissected out to analyze the distribution of axonal tracts and terminals of Brn3a-persistent neurons. tdTomato-positive cells were distributed in the lumbar spinal dorsal horn on the right. (B) Dorsal view of the spinal cord of AAV-injected mice. Solid and broken lines indicate the outline and the center of the spinal cord, respectively. (C) A transverse section of the lumbar spinal cord. A solid line indicates the outline of the spinal cord whereas a broken line indicates the boundary between the gray and white matters. Scale, 500 μm. (D) Distribution of

tdTomato (left) and SypEGFP (right) fluorescence throughout the spinal cord of AAV-injected mouse. The area where tdTomato-positive cells were distributed was about 1.5 mm along the rostrocaudal axis. Fluorescence image of the representative transverse section in this area is shown in "injection site". Fluorescence images rostrally (+2.0 cm, +1.5 cm, +1.0 cm, and +0.5 cm) and caudally (-0.5 cm and -1.0 cm) away from the injection site are also shown. The axonal tract labeled with tdTomato is distributed in the ipsilateral dorsal funiculus (DF), ipsilateral dorsolateral funiculus (DLF), ipsilateral ventrolateral funiculus (VLF), contralateral ventral funiculus (VF), contralateral VLF, or ipsilateral deep dorsal horn (circled by dotted magenta line). The fluorescence of tdTomato and SypEGFP is pseudocolored in black. The axonal tract on the contralateral side was mainly distributed in the VF near the injection site (+0.5 cm), but was found in the VLF on the section distant from the injection site (+2.0 cm, +1.5 cm, and +1.0 cm). Throughout the spinal cord, accumulation of presynaptic terminal of Brn3a-persistent neurons labeled with SypEGFP was found in the ipsilateral deep dorsal horn 0.5 cm and 1.0 cm rostral to the injection site (dotted green line). This region roughly corresponds to the Clarke nucleus in the thoracic spinal cord. Solid red lines indicate the outline of the spinal cord, whereas dotted red lines indicate the boundary between gray and white matters. Data are representative of images from 2 mice. (E-M) The distribution of presynaptic terminals of Brn3a-persistent spinal dorsal horn neurons in the medulla (E-I) and pons (J-M). (E) Schematic of the medulla with the location of nuclei (dotted red line) are shown. GN, gracile nucleus; CuN, cuneate nucleus; NTS, nucleus of the solitary tract; LRN, lateral reticular nucleus; IO, inferior olivary nucleus. Fluorescence images of SypEGFP (marked in E) around the dorsal (F, G) and ventral (H, I) medulla are shown. Strong EGFP signal was found in the ipsilateral GN, whereas weaker signal was also seen in the contralateral GN, and NTS, LRN, and IO on both sides. Solid and dotted red lines indicate the outline of the brain and the boundary of nuclei, respectively. (J) Schematic of the pons with the location of nuclei (dotted red line) is shown. Fluorescence images of SypEGFP (marked in J) around the dorsal (K, L) and ventral (M) pons are shown. EGFP signal was seen in the lateral parabrachial nucleus on both sides (arrows) and ventral area of the pons on the ipsilateral side. Red dotted lines indicate the outline of neuronal tracts and nuclei. scp, superior cerebellar peduncle; vsc, ventral spinocerebellar tract; un, uncinate fasciculus of the cerebellum; 7n, facial nerve; rs, rubrospinal tract; DPO, dorsal periolivary nucleus; LSO, lateral superior olive. Scale in F-I and K-M, 200 μm. Data are representative of images from 2 mice.
(PDF)

**S8 Fig. Schematics of axonal tracts and presynaptic targets of Brn3a-persistent neurons.** (A) Results in S8 Fig suggest that Brn3a-persistent neurons extend axons toward 5 different areas in the spinal cord: ipsilateral DF, ipsilateral dorsal horn, ipsilateral DLF, ipsilateral VLF and VF, and contralateral VLF and VF. The former 2 axons extend rostrally whereas the latter 3 axons do toward both rostral and caudal direction. (B) Previous anatomical studies suggest Brn3a-persistent neurons extending axons toward the ipsilateral DF innervate GN, whereas those extending axons toward LF and VF do several nuclei including NTS, LRN, or LPb. It is likely that Brn3a-persistent neurons extending axons toward ipsilateral deep dorsal horn innervate the Clarke nucleus.
(PDF)

**S9 Fig. Effect of Brn3a deficiency on the axonal extension of Brn3a-persistent neurons.** Axonal extension of Brn3a hetero and KO spinal dorsal neurons shown in Fig 3 was examined in the other mouse examples. AP staining images in the spinal cord of 2 Brn3a$^{cKOAP/+}$ (A, B) and 2 Brn3a$^{cKOAP/+}$ (C, D) mice are shown. Cre recombinase was introduced into the thoracic

(A-C) and cervical (D) spinal cord on the right. Scale, 200 μm.
(PDF)

**S10 Fig.** Rostrocaudal axonal extension of Brn3a-overexpressed neurons (A) *pCAG-LSL-EGFP* together with *pCAG-Brn3a* were focally introduced into spinal dorsal horn neurons of Brn3a^Cre/+ mice at E12.5 by *in utero* electroporation (0.5 mm round electrode). The spinal cord of the mice was dissected out at E18.5 to analyze the rostrocaudal axonal extension of Brn3a-overexpressed neurons. (B) The number of EGFP-positive cell bodies in the spinal dorsal horn as well as EGFP-positive axons in the ipsilateral VLF and VF was analyzed on the serial transverse sections of the spinal cord of 2 mice (mouse #4 and #7). The number of EGFP-positive cell bodies (blue) and EGFP-positive axons (orange) on each section is shown. The left and right sides of each graph indicate the sections located rostral and caudal to the electroporation sites, respectively.
(PDF)

**S11 Fig. AP assay of spinal cord sections of Lbx1-Cre;Brn3a-cKOAP mice.** AP assay was performed on the transverse sections the spinal cord of Lbx1^Cre/+;Brn3a^cKOAP/+ (A) and Lbx1^Cre/+;Brn3a^cKOAP/cKOAP(B) at E18.5. Nonspecific recombination occurred in a population of DRG neurons (red arrows). Scale, 500 μm.
(PDF)

**S12 Fig. Birthdate analysis of Brn3a-persistent spinal dorsal horn neurons.** EdU was intraperitoneally injected into the pregnant C57BL/6 mice at E11.5 and E12.5, and the spinal cord of the mice was dissected at P21. EdU staining together with immunostaining with anti-Brn3a antibody were performed on the transverse sections of the thoracic spinal cord of the mice. The percentage of EdU-positive cells among Brn3a-persistent ones is shown (E11.5:P21, $48.2 \pm 2.3\%$, 697 cells [$n = 3$ mice]; E12.5:P21, $48.2 \pm 2.3\%$, 750 cells [$n = 3$ mice]). Horizontal bars indicate the median.
(PDF)

**S1 File. The percentage of AP-positive neurons localized in Zone 1–5.** Raw data described in Fig 3H are shown.
(XLSX)

**S2 File. The percentage of control and Brn3a-overexpressed neurons localized in Zone 1–5.** Raw data described in Fig 6D are shown.
(XLSX)

**S3 File. Axonal extension index of control and Brn3a-overexpressed neurons.** Raw data described in Fig 7 are shown.
(XLSX)

## Acknowledgments

We thank Drs. N. Funatsu, T. Hirata, A. Inoue, S. Ito, T. Katano, T. Kimura, and R. Suno for their technical advices; Drs. C. Birchmeier, H. Itoh, T. Katano, J. Miyazaki, F. Murakami, R. Natsume, T. Saito, Y. Tanabe, E. Turner, K. Yamauchi, and Y. Zhu for materials; and Drs. Y. Zhu and T. Katano for critical reading of the manuscript. We would like to thank Editage (www.editage.com) for English language editing.

## Author Contributions

**Conceptualization:** Kazuhiko Nishida.

**Data curation:** Kazuhiko Nishida.

**Formal analysis:** Kazuhiko Nishida.

**Funding acquisition:** Kazuhiko Nishida.

**Investigation:** Kazuhiko Nishida.

**Methodology:** Kazuhiko Nishida, Shinji Matsumura.

**Project administration:** Kazuhiko Nishida.

**Resources:** Hitoshi Uchida, Manabu Abe, Kenji Sakimura, Tudor Constantin Badea.

**Supervision:** Kazuhiko Nishida, Takuya Kobayashi.

**Writing – original draft:** Kazuhiko Nishida.

**Writing – review & editing:** Kazuhiko Nishida.

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
