## [Decision Letter · Decision Letter 0]

19 Jul 2023

PONE-D-23-11663Brn3a controls the soma localization and axonal extension patterns of developing spinal dorsal horn neuronsPLOS ONE

Dear Dr. Nishida,

Thank you for submitting your manuscript to PLOS ONE. After careful consideration, the reviewers found that your article is very interesting, but there are some minor points that need to be addressed before publication. Therefore, we invite you to submit a revised version of the manuscript that addresses the points raised during the review process.

We look forward to receiving your revised manuscript.

Kind regards,

Carlos Oliva, PhD

Academic Editor

PLOS ONE

Journal Requirements:

2. Please expand the acronym “JSPS” (as indicated in your financial disclosure) so that it states the name of your funders in full.

Reviewers' comments:

Reviewer's Responses to Questions

**Comments to the Author**

1. Is the manuscript technically sound, and do the data support the conclusions?

Reviewer #1: Yes

Reviewer #2: Yes

2. Has the statistical analysis been performed appropriately and rigorously? 

Reviewer #1: Yes

Reviewer #2: Yes

3. Have the authors made all data underlying the findings in their manuscript fully available?

Reviewer #1: Yes

Reviewer #2: Yes

4. Is the manuscript presented in an intelligible fashion and written in standard English?

Reviewer #1: Yes

Reviewer #2: Yes

5. Review Comments to the Author

Reviewer #1: This manuscript that addresses the question of the molecular and genetic mechanisms of dorsal spinal cord development in mice. The authors focus on the POU family transcription factor Brn3a (aka Pou4f1) and its role in the migration and axonal projections of dorsal horn neurons. This question is novel and pertinent to the recent interest in the molecular underpinnings of pain-processing neural circuits found in the dorsal horn. The authors show that Brn3a is transiently expressed by a population of neurons, via their lineage tracing using a Cre knockin mouse. In mice with a Brn3a knockout, neurons in the superficial dorsal horn are displaced into deeper layers. Overexpression of Brn3a displaces superficial neurons into deeper layers. Additionally, the authors study the projection patterns of the Brn3a KO and overexpressing neurons and find some interesting changes consistent with at least some of them being spinofugal projection neurons. Additional manipulations include in utero electroporation of Brn3a or Cre expressing plasmids.

The manuscript is very well written and illustrated. All the data are convincingly documented and quantified. The data presented support very well the authors’ conclusions. I commend the authors on their technical prowess, especially on the in utero electroporation of the embryonic spinal cord.

I only have minor concerns:

The supplemental data are very useful, complete and convincing, and the authors should be encouraged to include them in the main figures, except for the statistics tables.

Some evidence of the spinal cord-specificity of their in utero electroporation is welcome.

The authors should consider discussing the ideas put forth in:

Sagner, A. et al. A shared transcriptional code orchestrates temporal patterning of the central nervous system. Plos Biol 19, e3001450 (2021)

and

Sagner, A. & Briscoe, J. Establishing neuronal diversity in the spinal cord: a time and a place. Development 146, dev182154 (2019).

Reviewer #2: In efforts to identify new transcription factors that might define specific subtypes of spinal cord neurons, Nishida et al. sought out to analyze the role of Brn3a in the development of dorsal horn neurons. By combining immunohistochemistry and lineage tracing in genetically modified mice, the authors studied Brn3a expression pattern and dynamics. Brn3a is transiently expressed by a wide population of dorsal horn neurons during development, whereas it persists in only a small population of neurons that exhibit a distinct localization pattern. The authors’ functional analysis allowed them to conclude that Brn3a is required for the precise soma localization of neurons that express Brn3 into postnatal ages, while not required for the extension of their axons. By gain-of-function experiments, the authors show that overexpression of Brn3a in Brn3a-transient neurons is sufficient to counteract the natural Brn3a downregulation during development and its consequent decrease of Brn3a-expressing neuron numbers. Strikingly, Brn3a overexpression also induces the misexpressing cells to locate to the marginal zone and deeper layers of the spinal cord forming a pattern that resembles the Brn3a-void region. The axons of these neurons show similar trajectories to their naturally Brn3a-expressing counterparts. Together, these results show that Brn3a is sufficient to confer the basic features of Brn3a-persistent neurons.

Overall, the study is nicely executed, and the authors’ main conclusions are supported by the data. However, I identified some areas for improvement of the manuscript.

Important points:

1. Regarding methods, how were the spinal cords dissected out and embedded? They roof plate looks intact in the cross sections. Also, a diagram and axes scheme would help the reader have a better idea of what images in figure 1 are showing.

2. The interpretation in lines 327-329 indicating that “These results suggest that Brn3a-positive neurons localized in the marginal region are derived from those in deeper region, which migrate through the void region” needs to be re-worded in my opinion. I do not see the logic for this statement, as Brn3a-positive cells are present in the marginal region throughout the analyzed stages. At least some of those cells seem to have always been there. Are there other possibilities? I think the author’s statement is possible, but to me it just does not stand out as the only/most likely reason, as it is written. I think more data would be needed to suggest that interpretation.

3. I suggest the authors having table 1 as a graph at the end of figure 1 for easier visualization of the data.

4. In line 414, the conclusion should focus on the function of Brn3 and not on the phenotypes of Brn3a-KO mice. This conclusion also seems to assume that the lost Zfhx3-positive neurons would have been Brn3a-persistent ones, but there is no data to support that. That should be noted.

5. Since the N-terminal region of Brn3a seems to be required for the localization of Brn3a-persistent neurons to the marginal zone, it would be great to expand the discussion about the function of this domain and the possible molecular mechanisms in place in this case.

For clarification:

1. From abstract, does line 36 refer to Brn3a expression decreasing over time during embryonic development? I think its needs to be clarified. Also, maybe add that the Brn3a-persistent population continues to express Brn3a at high levels.

2. Can the authors consider referring to the Brn3a-persistent population of neurons a different way? At least for the instances when they are referred to after loss of Brn3a. It sounds counterintuitive to refer to Brn3a-persistent cells to cells that do not express Brn3a. Maybe something like “cells that would otherwise be Brn3a-persistent” or “the cells that would have been Brn3a-persistent”, for example?

3. I strongly suggest to change the wording when using the term “KO”. In many cases it can be replaced by “the loss of Brn3a” instead of Brn3a-KO. In my opinion, this slight change focuses the attention on the actual biological consequence rather than the technical approach. For example, the subheading for results part “Effect of Brn3a-KO on the distribution of Brn3a-persistent spinal dorsal horn neurons”, and other subheadings and text throughout.

4. In figure S2, the EGFP expression does not match fully that of Brn3a immunostain. Did the authors introduce a different plasmid as control to detect transfection efficiency independent of Cre activity? That could potentially explain the discrepancy in the signal.

Minor points:

1. Line 36 should say “was downregulated in the majority”

2. Line 219, “basic” instead of “besic”

3. Line 278, it should say pCAG instead of pCAL

4. Throughout the manuscript, change “SXX fig” to “Fig SXX”

5. Typo lumber, change to lumbar figure S6 legend

6. Line 1027, it should say pCAG instead of pCAL

6. PLOS authors have the option to publish the peer review history of their article (what does this mean?). If published, this will include your full peer review and any attached files.

Reviewer #1: No

Reviewer #2: No

---

## [Author Response · Author response to Decision Letter 0]

21 Aug 2023

We changed one sentence and one figure as indicated in our cover letter

(Change-1) Line 721: “Loss of Brn3a did not affect the overall axonal extension of Brn3a-persistent neurons toward the DF, LF, or VF (Fig 5).” The phrase “did not affect the overall axonal extension” used to be “did not affect overall the axonal extension” in the original manuscript.

(Change-2) S5 Figure: The gap between images was adjusted properly. 

Question（Editor-1） Please ensure that your manuscript meets PLOS ONE's style requirements, including those for file naming. The PLOS ONE style templates can be found at 

Answer: We believe our manuscript meets PLOS ONE’s style requirements.

Question（Editor-2） Please expand the acronym “JSPS” (as indicated in your financial disclosure) so that it states the name of your funders in full. This information should be included in your cover letter; we will change the online submission form on your behalf.

Answer: We described the funder as “Japan Society for the Promotion of Science” instead of “JSPS” in our cover letter.

Question（Editor-3） We note that you have included the phrase “data not shown” in your manuscript. Unfortunately, this does not meet our data sharing requirements. PLOS does not permit references to inaccessible data. We require that authors provide all relevant data within the paper, Supporting Information files, or in an acceptable, public repository. Please add a citation to support this phrase or upload the data that corresponds with these findings to a stable repository (such as Figshare or Dryad) and provide and URLs, DOIs, or accession numbers that may be used to access these data. Or, if the data are not a core part of the research being presented in your study, we ask that you remove the phrase that refers to these data.

Answer: As editor pointed out, we mentioned our preliminary data in discussion section without showing the data (line 589 in the original manuscript). We have now added this data (S11 Fig) and figure legend (Line 1118-1121) in the revised manuscript.

Question（Editor-4） Please review your reference list to ensure that it is complete and correct. If you have cited papers that have been retracted, please include the rationale for doing so in the manuscript text, or remove these references and replace them with relevant current references. Any changes to the reference list should be mentioned in the rebuttal letter that accompanies your revised manuscript. If you need to cite a retracted article, indicate the article’s retracted status in the References list and also include a citation and full reference for the retraction notice.

Answer: None of the references we cited have been retracted. In the revised manuscript, we added 8 additional references (Reference 17 [Line 836-840]: Zhu et al, Reference 36 [Line 905-909]: Sathyamurthy et al, Reference 37 [Line 910-913]: Haring et al, Reference 38 [Line 914-916]: Sagner and Briscoe, Reference 41 [Line 923-926]: Smith et al, Reference 42 [Line 927-930]: Smith et al, reference 43: Smith et al [Line 931-935], Reference 44 [Line 936-939]: Sugars et al) according to comments from reviewers #1 and #2.

Question（Reviewer #1-1） The supplemental data are very useful, complete and convincing, and the authors should be encouraged to include them in the main figures, except for the statistics tables.

Answer: According to reviewer #1’s comment, we included S2 figure and S6 figure (shown in the original manuscript) as main figures in the revised version, and they became to be Fig 2 and Fig 4, respectively. The number of Zfhx3-positive neurons in S2 figure panel in the original manuscript was removed in the revised manuscript (Fig 4) because it was shown in the main text already (line 426). The rest of the supplementary figures are supportive data in our opinion and are not directly involved in the development Brn3a-positive neurons. Thus, we decided to leave them in supplemental figures in the revised manuscript.

Question (Reviewer #1-2) Some evidence of the spinal cord-specificity of their in utero electroporation is welcome.

Answer: According to reviewer #1’s comment, we added S2 Figure in the revised manuscript which shows spinal dorsal horn-specific gene transfer by in utero electroporation. In this experiment, we introduced pCAG-mCherry into E12.5 spinal cord and the spinal cord sample was dissected at E18.5. As shown in the figure, mCherry-positive cells were only found in the spinal dorsal horn on the right (electroporated side) but not in the DRG, demonstrating that this method enables spinal dorsal horn-specific gene transfer. We described pCAG-mCherry vector in Method and Acknowledgement in the revised manuscript (line 117, 120, and 770).

Question (Reviewer #1-3) The authors should consider discussing the ideas put forth in: 

Sagner, A. et al. A shared transcriptional code orchestrates temporal patterning of the central nervous system. Plos Biol 19, e3001450 (2021)

and

Sagner, A. & Briscoe, J. Establishing neuronal diversity in the spinal cord: a time and a place. Development 146, dev182154 (2019).

Answer: Reviewer #1 pointed out that we did not discuss enough about cell fate specification of Brn3a-transient and consistent neurons in the discussion section in the original manuscript. We added an additional subsection in the discussion (Cell fate specification of Brn3a transient and persistent neurons) of the revised manuscript, and discussed this issue from the point of view of spatial and temporal patterning of spinal development (line 632-651).

Question (Reviewer #2-1) Regarding methods, how were the spinal cords dissected out and embedded? They roof plate looks intact in the cross sections. Also, a diagram and axes scheme would help the reader have a better idea of what images in figure 1 are showing.

Answer: According to reviewer #2’s comment, we described our dissection process in more detail in Method in the revised manuscript (Line 144-149), in which “Embryonic mice from E14.5 to E18.5 were transcardially perfused with 4% paraformaldehyde solution (0.1 M phosphate buffer [pH 7.4] containing 4% paraformaldehyde) whereas those at E12.5 were dissected without transcardial perfusion. Spinal cords at E16.5 and E18.5 were then peeled off from the vertebral column whereas those from E12.5 and E14.5 were kept with it in order not to damage the sample during dissection.” In terms of embedding process, we believe we have provided enough information in Method in the original manuscript already. In addition, we added diagrams of the spinal cord in the right bottom corner of Fig 1A-D to show that spinal cord images shown in Fig 1A-D indicate the dorsal spinal cord on the right. We described it in the Figure legend accordingly (line 348-350). In addition, the spinal cord images shown in Fig 1H-J in the original manuscript were not within the right frame (including the right edge of the spinal cord and DRG), which may confuse readers. In the revised manuscript, the original images were replaced by those of the same sections including the right half of the dorsal horn together with the midline. 

Question (Reviewer #2-2) The interpretation in lines 327-329 indicating that “These results suggest that Brn3a-positive neurons localized in the marginal region are derived from those in deeper region, which migrate through the void region” needs to be re-worded in my opinion. I do not see the logic for this statement, as Brn3a-positive cells are present in the marginal region throughout the analyzed stages. At least some of those cells seem to have always been there. Are there other possibilities? I think the author’s statement is possible, but to me it just does not stand out as the only/most likely reason, as it is written. I think more data would be needed to suggest that interpretation.

Answer: We agree with reviewer #2 in that the conclusion written in lines 327-329 in the original manuscript is only one possibility inferred from Brn3a expression pattern of developing spinal cord (Fig 1) and we think it is too early to discuss about the migration of Brn3a-positive neurons. Since we discussed the role of Brn3a in the localization of Brn3a-persistent neurons in the discussion section, we removed the description in lines 327-329 (written in the original manuscript) in the revised manuscript.

Question (Reviewer #2-3) I suggest the authors having table 1 as a graph at the end of figure 1 for easier visualization of the data.

Answer: According to reviewer #2’s comment we showed these data (Table 1 in the original manuscript) in Fig 1K and 1L as bar graphs in the revised manuscript. We described them in Figure legends (line 352-355) and main text (line 332 and 377) in the revised manuscript.

Question (Reviewer #2-4) In line 414, the conclusion should focus on the function of Brn3 and not on the phenotypes of Brn3a-KO mice. This conclusion also seems to assume that the lost Zfhx3-positive neurons would have been Brn3a-persistent ones, but there is no data to support that. That should be noted.

Answer: Following reviewer #2’s comment, we focused on the function of Brn3a (and Brn3b) in the laminar localization of Brn3a-persistent neurons instead of Brn3a-KO phenotype in the revised manuscript. The revised discussion is in line 427-430 which states “These results suggest that Brn3a plays critical roles in the localization of Brn3a-persistent neurons in the marginal laminae, and that Brn3b is functionally equivalent to Brn3a in the developmental process.”

Question (Reviewer #2-5) Since the N-terminal region of Brn3a seems to be required for the localization of Brn3a-persistent neurons to the marginal zone, it would be great to expand the discussion about the function of this domain and the possible molecular mechanisms in place in this case.

Answer: According to reviewer #2’s comment, we expanded the discussion in this part in discussion section in the revised manuscript (line 671-690). We cited 4 additional references in the discussion.

Question (Reviewer #2-6) From abstract, does line 36 refer to Brn3a expression decreasing over time during embryonic development? I think its needs to be clarified. Also, maybe add that the Brn3a-persistent population continues to express Brn3a at high levels.

Answer: According to reviewer #2’s comment, we changed the description of line 36-38 in Abstract in the revised manuscript in which, “The majority of the Brn3a-lineage neurons ceased Brn3a expression during embryonic stages (Brn3a-transient neurons), whereas a limited population of them continued to express Brn3a at high levels after E18.5 (Brn3a-persistent neurons).”

Question (Reviewer #2-7) Can the authors consider referring to the Brn3a-persistent population of neurons a different way? At least for the instances when they are referred to after loss of Brn3a. It sounds counterintuitive to refer to Brn3a-persistent cells to cells that do not express Brn3a. Maybe something like “cells that would otherwise be Brn3a-persistent” or “the cells that would have been Brn3a-persistent”, for example?

Answer: We called this neuronal population as “Brn3a-persistent neurons” in Brn3a-KO situation in 6 parts in the main text (Line 39, 383, 484, 604, 724, 762), such as “Loss of Brn3a disrupted the localization pattern of Brn3a-persistent neurons…” (Line 38-39). Although terms such as “cells that would otherwise be Brn3a-persistent” and “the cells that would have been Brn3a-persistent” are logically more precise, they sound complicated in our opinion. We believe it is reasonable to use the word “Brn3a-persistent neurons” as in the abstract (Line 39), discussion (Line 704, 724, 762), and subtitle of result section (Line 383, 484). However, to help readers to understand this point clearly, we modified the result section in which the phenotype of Brn3aAP/AP neurons were described. We added one sentence just after the result of localization of Brn3aAP/AP neurons (line 412-414) in which “These results indicate that loss of Brn3a disrupts the localization pattern of Brn3a-persistent neurons in the spinal dorsal horn.” In the same way, we modified the result section in the end of the result of axonal extension of Brn3aAP/AP neurons (line 514-516) in which “These results suggest that loss of Brn3a does not affect overall axonal extension of Brn3a-persistent neurons toward the ipsilateral DF, LF, and contralateral VF.”

Question (Reviewer #2-8) I strongly suggest to change the wording when using the term “KO”. In many cases it can be replaced by “the loss of Brn3a” instead of Brn3a-KO. In my opinion, this slight change focuses the attention on the actual biological consequence rather than the technical approach. For example, the subheading for results part “Effect of Brn3a-KO on the distribution of Brn3a-persistent spinal dorsal horn neurons”, and other subheadings and text throughout.

Answer: According to reviewer #2’s comments, “Brn3a-KO” was replaced by “loss of Brn3a” (Line 38, 382, 481, 601, 725, 763), “Brn3a deficiency” (Line 87, 90, 386, 420, 432, 474, 483, 518, 663-664, 752, 1100) or “Brn3aAP/AP” (Line 397, 398, 407, 409-411, 414-418, 446-449, 452-459, 476, 655, 668, and Figure 3H) in the revised manuscript. Brn3a-hetero was also replaced by Brn3aAP/+ in the revised manuscript (Line 102, 392, 394, 408-411, 445, 451-452, 455-458, 656, and Fig 3H) 

Because of this change, we changed the subtitle of Result section (Loss of Brn3a disrupts the localization pattern of Brn3a-persistent neurons in the spinal dorsal horn neurons [Line 382-383], Loss of Brn3a did not affect overall axonal extension of Brn3a-persistent neurons [Line 481-482], Brn3a overexpression directs the soma localization in a manner similar to Brn3a-persistent neurons [Line 526-527], Brn3a overexpression directs the axonal extension in a manner similar to Brn3a-persistent neurons [Line 567-568]). 

Question (Reviewer #2-9) In figure S2, the EGFP expression does not match fully that of Brn3a immunostain. Did the authors introduce a different plasmid as control to detect transfection efficiency independent of Cre activity? That could potentially explain the discrepancy in the signal.

Answer: The reason why EGFP expression did not match that of Brn3a immunostaining is because we introduced EGFP expression vector (pCAL-LSL-EGFP) at E12.5 when most Brn3a-persistent neurons exit cell cycle. As shown in Fig S11, about 20% of Brn3a-persistent neurons exit final cell cycle after E12.5. The percentage almost corresponds to the percentage of Brn3a positive cells among EGFP positive cells shown in Fig 2. 

Question (Reviewer #2-10) Line 36 should say “was downregulated in the majority”

Answer: We appreciated the comment. We corrected it accordingly (Line 36 in the revised manuscript).

Question (Reviewer #2-11) Line 219, “basic” instead of “besic”

Answer: We appreciated the comment. We corrected it accordingly (Line 222 in the revised manuscript).

Question (Reviewer #2-12) Line 278, it should say pCAG instead of pCAL

Answer: We appreciated the comment. We corrected it accordingly (line 281 in the revised manuscript).

Question (Reviewer #2-13) Throughout the manuscript, change “SXX fig” to “Fig SXX”

Answer: We also prefer “Fig SXX” style but we are supposed to write “SXX Fig” according to the journal guideline (https://journals.plos.org/plosone/s/file?id=wjVg/PLOSOne_formatting_sample_main_body.pdf.). 

Question (Reviewer #2-14) Typo lumber, change to lumbar figure S6 legend

Answer: We appreciated the comment. According to Reviewer #1’s comment, Fig S6 became to be Fig 4 in the revised manuscript. We corrected them accordingly (Line 464 and 467 in the revised manuscript). 

Question (Reviewer #2-15) Line 1027, it should say pCAG instead of pCAL

Answer: We appreciated the comment. We corrected it accordingly (Line 1107 in the revised manuscript).

---

## [Decision Letter · Decision Letter 1]

10 Sep 2023

Brn3a controls the soma localization and axonal extension patterns of developing spinal dorsal horn neurons

PONE-D-23-11663R1

Dear Dr. Nishida,

We’re pleased to inform you that your manuscript has been judged scientifically suitable for publication and will be formally accepted for publication once it meets all outstanding technical requirements.

Kind regards,

Carlos Oliva, PhD

Academic Editor

PLOS ONE

Additional Editor Comments (optional):

Reviewers' comments:

Reviewer's Responses to Questions

**Comments to the Author**

1. If the authors have adequately addressed your comments raised in a previous round of review and you feel that this manuscript is now acceptable for publication, you may indicate that here to bypass the “Comments to the Author” section, enter your conflict of interest statement in the “Confidential to Editor” section, and submit your "Accept" recommendation.

Reviewer #1: All comments have been addressed

Reviewer #2: All comments have been addressed

2. Is the manuscript technically sound, and do the data support the conclusions?

Reviewer #1: Yes

Reviewer #2: Yes

3. Has the statistical analysis been performed appropriately and rigorously? 

Reviewer #1: Yes

Reviewer #2: Yes

4. Have the authors made all data underlying the findings in their manuscript fully available?

Reviewer #1: Yes

Reviewer #2: Yes

5. Is the manuscript presented in an intelligible fashion and written in standard English?

Reviewer #1: Yes

Reviewer #2: Yes

6. Review Comments to the Author

Reviewer #1: This is very nice work, the authors did a terrific job with the revisions. The range of technical approaches is impressive.

Reviewer #2: The authors have addressed all my comments and requests for clarification. I think these changes make the paper stronger, more accurate and easier to read.

I humbly apologize for requesting the change of supplementary figure naming, I realized after submitting my review that that was a journal's requirement.

I think this is a very nice story, strongly supported and it deserves to be published.

7. PLOS authors have the option to publish the peer review history of their article (what does this mean?). If published, this will include your full peer review and any attached files.

Reviewer #1: No

Reviewer #2: No

---

## [Editor Report · Acceptance letter]

14 Sep 2023

PONE-D-23-11663R1 

Brn3a controls the soma localization and axonal extension patterns of developing spinal dorsal horn neurons 

Dear Dr. Nishida:

I'm pleased to inform you that your manuscript has been deemed suitable for publication in PLOS ONE. Congratulations! Your manuscript is now with our production department. 

Kind regards, 

on behalf of

Dr. Carlos Oliva 

Academic Editor

PLOS ONE